# Land surface model performance using cosmic-ray and point scale soil moisture measurements for calibration

Joost Iwema[1], Rafael Rosolem[1,2], Mostaquimur Rahman[1], Eleanor Blyth[3], Thorsten Wagener[1,2]

[1]Department of Civil Engineering, University of Bristol, Bristol, UK
[2]Cabot Institute, University of Bristol, Bristol, UK
[3]Centre for Ecology and Hydrology, Wallingford, OX10 8BB, UK

*Correspondence to*: Joost Iwema (joost.iwema@bristol.ac.uk)

**Abstract.** At very high resolution scale (i.e. grid cells of 1 km$^2$) land surface model parameters can be calibrated with Eddy-Covariance flux data and point-scale soil moisture data. However, measurement scales of Eddy-Covariance and point-scale
data differ substantially. In our study, we investigated the impact of reducing the scale mismatch between surface energy flux and soil moisture observations by replacing point-scale soil moisture data with observations derived from Cosmic-Ray Neutron Sensors made at larger spatial scales. Five soil and evapotranspiration parameters of the Joint UK Land Environment Simulator (JULES) were calibrated against point-scale and Cosmic-Ray Neutron Sensor soil moisture data separately. We calibrated the model for twelve sites in the USA representing a range of climatic, soil, and vegetation conditions. The improvement in latent
heat flux estimation for the two calibration solutions was assessed by comparing to Eddy-Covariance flux data and to JULES simulations with default parameter values. Calibrations against the two soil moisture products alone did show an advantage for the Cosmic-Ray technique. However, further analyses of two-objective calibrations with soil moisture and latent heat flux showed no substantial differences between both calibration strategies. This was mainly caused by the limited effect of calibrating soil parameters on soil moisture dynamics and surface energy fluxes. Other factors that played a role were limited
spatial variability in surface fluxes implied by soil moisture spatio-temporal stability, and data quality issues.

## 1 Introduction

The land surface water and energy balances are coupled through the process of evapotranspiration. Soil moisture is one of the main water reservoirs near the land surface and can hence importantly control the surface water and energy balances. Soil moisture provides a first order (i.e. direct) control on evapotranspiration when there is insufficient water to meet the evaporative
demand (Manabe et al., 1969; Budyko, 1956; Seneviratne et al., 2010). An indirect effect of soil moisture on surface energy flux partitioning is for instance the damping effect on soil and air temperature, which in its turn affects humidity, evapotranspiration, boundary-layer stability, and in some cases precipitation (Seneviratne et al., 2010). The control of soil moisture on temperature at seasonal scales is especially strong in transitional climate regions (Koster et al., 2004).

Land Surface Models (LSMs) solve the surface mass (including water), energy, and momentum balances to provide the weather
and climate prediction models with lower boundary conditions. The land surface has been shown to play an important role in

global atmospheric circulation (Koster et al., 2004). Because the soil moisture state and surface fluxes are so closely connected, it is important to accurately simulate these simultaneously (Henderson-Sellers et al., 1996; Richter et al., 2004; Seneviratne et al., 2010; Dirmeyer, 2011; Dirmeyer et al., 2013).

The increasing complexity of land surface models over the past decades (Sellers, 1997; Seneviratne et al., 2010) has brought
with it an increasing number of parameters, with values not easily defined with in-situ measurements because of scale mismatch. For instance, stomatal resistance measured at leaf level is not the same as canopy stomatal resistance needed for LSMs (Blyth et al., 1993). Soil hydraulic parameter values (e.g. soil hydraulic conductivity) are often obtained from laboratory experiments on soil cores for cubic centimetres to cubic decimetres. The soil properties and processes at this scale can however differ from those at the LSM grid cell sizes, which are often as large as hundreds to tens of thousands of square kilometres
(Pitman 2003). Due to the different governing processes upscaling the soil hydraulic properties from soil core scale to field scale is non-trivial (Vinnikov 1996; Crow et al., 2012).

In an effort to develop global hydrometeorological monitoring and prediction capabilities (Wood et al., 2011), hydrological models and LSMs are now increasingly being applied at the finer `hyper-resolution scale' with grid cells of about 1 km$^2$. Typically, parameters are calibrated and validated at this hyper-resolution against in-situ measurements from sources such as
eddy-covariance flux towers and point-scale (PS) soil moisture sensors (e.g. Time Domain Transmissivity; and Time Domain Reflectometry) (Stockli et al., 2008; Richter et al., 2004; Blyth et al., 2010; Blyth et al., 2011; Rosolem et al., 2013). Such calibration/validation data has become more widely available at hyperresolution scale (Baldocchi, 2001; Smith, 2012). However, the horizontal footprints of different measurement techniques' vary from each other: Eddy-covariance surface energy flux data represent a downwind footprint of 100 m$^2$ to 1 km$^2$, while in-situ soil moisture probes link to much smaller surface
areas by representing a support volume (Blöschl and Sivapalan, 1995) of ~4 dm$^3$ only (Running et al., 1999; Kurc and Small, 2007; Vivoni et al., 2008). Soil moisture is spatially non-uniform within the eddy-covariance footprint due to heterogeneity in soil properties, vegetation, and topography. Therefore, soil moisture measurements best (i.e. most effectively) representing the soil below the eddy-covariance tower's footprint should be used when the performance of a land surface model is evaluated. If soil moisture is measured at only a single or few locations with limited support volume, like with PS sensors, the measured
soil moisture content might be different from the effective soil moisture state that controls the surface exchange processes. It is therefore often assumed that soil moisture measured at a scale closer to the footprint of an eddy-covariance tower is more informative than a single or a couple of PS sensor profiles for studying land-surface processes and constraining model parameters at scales of >~100 m$^2$ (Robinson et al., 2008). This poses a potential scale mismatch issue when a single or a few PS sensors are used. On the other hand, past research has shown soil moisture measured at only one or a few points within an
area of similar size as an eddy-covariance footprint, can have similar value to surface energy flux simulation as soil moisture measured at a larger scale (e.g. Vachaud et al., 1985; Teuling et al., 2006; Mittelbach and Seneviratne, 2012). These studies showed that points within a soil moisture observation network keep their rank with respect to the mean soil moisture (anomaly), i.e. they either under –or overestimate the mean (anomaly), so-called spatio-temporal stability. The physical principle behind the spatio-temporal stability theory is that different time variant and hydrological processes either create or destroy spatial soil

moisture variability whereas time invariant land surface characteristics induce an effective offset in the spatial variability (Albertson and Montaldo, 2003; Teuling and Troch, 2005; Vanderlinden et al., 2012). When soil moisture reaches values below the critical point (i.e. transpiration becomes moisture limited), spatial variability in soil moisture and fluxes is reduced (Teuling and Troch, 2005). Soil moisture dynamics was found to be a small portion of total soil moisture variability (Mittelbach

and Seneviratne, 2012), while having a greater effect on surface energy fluxes than absolute soil moisture in land surface models (Dirmeyer et al., 1999; Teuling et al., 2009). The implication of the spatio-temporal stability theory therefore is that the spatial scale mismatch issue might have limited implications to surface energy flux simulation. The question which then rises is,

*does reduced observation scale mismatch improve LSM energy flux estimates?*

Based on the spatio-temporal stability theory, we phrased the following hypothesis for our research question:

*reduced observation scale mismatch does not lead to LSM energy fluxes closer to eddy covariance observations.*

In recent years, new soil moisture measurement techniques have been developed that have, compared to point-scale soil moisture sensing techniques, a reduced scale mismatch with eddy-covariance surface energy flux measurements. Improvement

in wireless technology and remote data collection technology have made the development of PS soil moisture sensor networks more feasible (Cardell-Oliver, 2005; Ritsema et al., 2009; Trubilowicz, 2009; Bogena et al., 2010; Robinson et al., 2008). Newer soil moisture sensor techniques, for instance one which makes use of the Global Positioning System (GPS; Larson et al 2008, 2010), and the Cosmic-Ray Neutron Sensor (CRNS; Zreda et al., 2008) have the advantage that their installation requires less time and work effort because only a single above ground sensor is needed. The CRNS (Zreda et al., 2008) is an

above-ground passive sensor which utilises natural cosmic-ray neutron radiation to estimate soil moisture content in the top 10-70 cm. The sensor's footprint area has a radius of about 100 to 300 m surrounding the above-ground sensor (Desilets and Zreda, 2013; Köhli et al., 2015). Franz et al. (2012) showed soil moisture estimated from CRNS neutron counts differed less than 20% from the average of a co-located point scale soil moisture sensor network at a site in Arizona. Networks of CRNS have been established in various countries (e.g. Cosmic-ray Soil Moisture Observation System; COSMOS (Zreda et al., 2012),

COSMOS-UK (Evans et al., 2016), the Australian National Cosmic-Ray Soil Moisture Monitoring Facility CosmOZ (Hawdon et al., 2004), and TERrestrial ENvironmental Observatoria; TERENO (Baatz et al., 2015)). Unlike wireless point scale sensor networks, both the GPS and CRNS technology provide an integrated soil moisture measurement over the entire support volume (Larson et al., 2008; Zreda et al., 2008). We chose to answer our research question using the CRNS technology because the COSMOS network provides publicly available data for multiple years at a range of

sites co-located with AmeriFlux/FLUXNET eddy-covariance towers (ORNL-DAAC, 2015). Twelve of these sites provided sufficient LSM forcing data, PS soil moisture data, CRNS data, and eddy-covariance LE and sensible heat flux (H) data.

Before our modelling exercise we first compared the PS and CRNS data. The outcomes of this data analysis were mainly used to see if the results from the calibration and validation yielded larger differences in surface energy flux estimation at sites

where the two soil moisture observation products showed higher deviation from each other. To investigate our research question we made the LSM simulated soil moisture content match the observed PS or CRNS data as closely as possible. We did this by calibrating parameters of the Joint UK Land Environment Simulator (JULES; Best et al., 2011) against point-scale and Cosmic-Ray Neutron data separately. We subsequently validated the results against eddy-covariance observed data over

the same periods. To assess the change in soil moisture and surface energy fluxes after calibration we compared the calibrated runs against a default run with parameter values computed from a widely used soil properties database. We emphasise that we compared the two different soil moisture measurement techniques' scales and not the techniques as such.

## 2 Data and Methods

### 2.1 Calibration and Validation data: PS, CRNS, and eddy-covariance data

Point scale (PS) soil moisture and CRNS neutron count data from twelve AmeriFlux/COSMOS sites were used (Figure 1; full COSMOS site names are shown in this figure). These twelve sites covered eight of twenty Ecoclimatic domains of the US National Ecological Observatory Network (NEON; www.neonscience.org) (Figure 1). These twelve sites hence represent a variety of climates and land cover types, but also different soil types (Table 1).

Hourly PS data for nine sites were obtained from the publicly available AmeriFlux Level 2 data source (ORNL). Data for the

three California Climate Gradient sites (DC, CS, and SO) were obtained at http://www.ess.uci.edu/~california/ (data version 3.4; Goulden et al., 2015). The number of PS profiles, the installation depths, and sensor types differed between the twelve study sites (Table A1.1 in Appendix 1). We used point scale soil moisture data from the soil layers up to 30 cm depth only for consistency among all sites. There were only two sites reporting soil moisture data at greater depths: WR at 50 cm, and MO at 100 cm). Our main objective was to investigate the difference in information content due to two soil moisture measurement

techniques' different horizontal scales in relation to the eddy-covariance footprint, rather than to compare the measurement techniques themselves. Quality control was applied to filter out spurious and unrealistic data points due to sensor errors. The PS data was then interpolated to the JULES soil layer on which the model was calibrated.

Hourly CRNS neutron count data were obtained from the COSMOS network website (www.cosmos.hwr.arizona.edu). Corrections were applied as by Zreda et al. (2012). Water vapour corrections (Rosolem et al., 2013) were applied with respect

to dry air (Bogena et al., 2013). The quality control approach used for the PS analysis was also applied to CRNS neutron count data series to remove unrealistic points. Snow cover periods were also removed for the analysis. A 5-hour moving average window was applied to the observed CRNS neutron counts (following Shuttleworth et al., 2013; Rosolem et al., 2014).

We used the exact same data for the soil moisture data comparison as for the calibration (providing that both PS and CRNS were available in the same period). As validation data, latent heat (LE) flux hourly data from AmeriFlux Level 2 data source

was used for nine sites. We used data version 3.4 (Goulden et al., 2015) for the three California Climate Gradient sites. Quality control was applied to the LE and H flux data to remove outliers and unrealistic data points.

## 2.2 Soil moisture data comparison methodology

To compare PS and CRNS soil moisture data we computed the Mean Squared Deviation (MSD) and its decomposition (Gupta et al., 2009) into:

(1) The squared difference between the means (structural bias)

(2) The squared difference between the standard deviations (indicates different seasonality)

(3) A term relating to the coefficient of linear correlation (indicates different dynamics)

Which yields the following equation:

$$MSD = (\mu_{PS} - \mu_{CRNS})^2 + (\sigma_{PS} - \sigma_{CRNS})^2 + 2 \cdot \sigma_{PS} - \sigma_{CRNS} \cdot (1 - r)$$

Where $\mu$ ($m^3m^{-3}$) is the observed mean, $\sigma$ ($m^3m^{-3}$) is standard deviation, and r (-) is the coefficient of linear correlation. We however scaled the relative contributions of the three MSD components to the Root Mean Squared Deviation (RMSD) instead of MSD, to keep units of $m^3m^{-3}$. We then ranked the sites from lowest to highest RMSD. According to our hypothesis, we would expect to see a larger difference in simulated surface energy fluxes after calibration when the two soil moisture time series differ most. We compared PS soil moisture with CRNS soil moisture values computed from vertically homogeneous soil moisture values obtained from the observed neutron counts using the COsmic-ray SoIl Moisture Interaction Code (COSMIC; Shuttleworth et al., 2013).

## 2.3 JULES forcing data and initial conditions

JULES requires precipitation, air temperature, atmospheric pressure, wind speed, specific humidity, downward shortwave radiation, and downward longwave radiation as meteorological forcing data. Quality controlled hourly data was obtained from AmeriFlux Level 2 and the three California Climate Gradient sites. At some of the sites however, certain specific forcing data was not available from AmeriFlux and hence data from different sources were used (Table A1.2 of Appendix 1).

Model input data was gap-filled following Rosolem et al. (2010) because JULES requires continuous time series except for precipitation where gaps were set to zero. For all sites gaps smaller than 3 hours were filled by linear interpolation. Larger gaps of up to 30 days were gap-filled using the average diurnal pattern of the preceding and following 15 days. In addition, some remaining gaps in Downward Shortwave Radiation and Downward Longwave Radiation at Wind River (WR) were filled using linear least squares relationships with NLDAS-2 data (http://disc.sci.gsfc.nasa.gov/uui/datasets?keywords=NLDAS). At Soaproot (SO) and Coastal Sage (CS) data gaps in the atmospheric pressure time series were filled with NLDAS data. At CS NLDAS data were also used to gap fill air temperature. Gap filling at Santa Rita Creosote was done with data from the nearby Sahuarita site followed by the gap-filling procedure described above.

## 2.4 Calibration and validation methodology

### 2.4.1 Joint UK Land Environment Simulator (JULES)

We used the Joint UK Land Environment Simulator (JULES; Best et al., 2011; Clark et al., 2011) in this study. JULES can be coupled as lower boundary condition to the UK Met Office Unified Model (Cullen, 1993). Within JULES choices can be made
(e.g. canopy radiation model type) and certain modules (e.g. vegetation dynamics) can be switched on or off to operate at different levels of complexity. In addition, we chose the UK Variable resolution configuration (UKV) because it is the standard setting when JULES is run coupled with the UK Met Office Unified Model. The UKV land grid cell size is 1 km by 1 km. However, our study focused on JULES standalone simulations at the 12 grid points located at the sites investigated. The UKV setting employs the multi-layer canopy radiation module with surface heat capacity and snow beneath the canopy, the single
canopy layer `big leaf' approach for leaf-level photosynthesis (which computes radiation absorption with Beer's law). Soil heat conductivity was calculated using the approach of Dharssi et al. (2009). We used the default JULES-UKV soil layering (Supplement 1, Figure S1.1). The hydraulic bottom boundary condition in JULES is free drainage.

JULES computes the transport of water through the soil using a finite difference representation of the Richards' equation. The vertical fluxes are computed with the Buckingham-Darcy equation. JULES-UKV uses the Mualem-Van genuchten (Van
Genuchten, 1980; Mualem, 1983) soil water retention equations. The Van Genuchten equation calculates soil water content $\theta$ ($m^3 m^{-3}$) from soil hydraulic pressure head $\psi$ (m):

$$\frac{\theta - \theta_{res}}{\theta_{sat} - \theta_{res}} = \frac{1}{[1+(\alpha\Psi)^n]^{1-1/n}}, \tag{1}$$

with shape parameter n (-), $\alpha$ ($m^{-1}$) representing the inverse of the water entry pressure, $\theta_{res}$ ($m^3 m^{-3}$) is the empirical residual soil moisture content (without physical meaning), and $\theta_{sat}$ (or smsat; $m^3 m^{-3}$) is the saturated soil moisture content. In JULES
parameter n is defined as b = 1/(n-1) and sathh=$\alpha^{-1}$ (m).

The Mualem equation computes the unsaturated hydraulic conductivity K:

$$K = K_{sat} \frac{\theta - \theta_{res}}{\theta_{sat} - \theta_{res}} \left[ 1 - (1 - \frac{\theta - \theta_{res}}{\theta_{sat} - \theta_{res}}^{1/(1-\frac{1}{n})})^{1-\frac{1}{n}} \right]^2, \tag{2}$$

where $K_{sat}$ (or satcon; $mm \cdot s^{-1}$) is the saturated hydraulic water conductivity.

The values of the Mualem-Van Genuchten parameters need to be defined by the user for each grid cell/point based on soil
characteristics.

In JULES soil moisture directly interacts with transpiration (through root water uptake) and bare soil evaporation as described hereafter (see also Figure S1.2 of Supplement 1). JULES first computes the potential photosynthesis, which is a function of three limiting factors: Rubisco limitation, radiation limitation, and photosynthetic product transport limitation. The potential photosynthesis is multiplied with a soil moisture reduction factor to obtain the actual photosynthesis. To obtain this soil
moisture reduction factor the model first computes a limiting factor for each layer:

$$\beta_i = \begin{cases} 1, & \theta_i \geq \theta_{crit} \\ \frac{(\theta_i - \theta_{wilt})}{(\theta_{crit} - \theta_{wilt})}, & \theta_{wilt} < \theta_i < \theta_{crit}, \\ 0, & x \leq \theta_{wilt} \end{cases} \tag{3}$$

where $\theta_i$ is the unfrozen soil moisture content in layer $i$, $\theta_{crit}$ is the critical point soil moisture content below which soil moisture is limiting the root water uptake (matrix water potential -330 cm in JULES), and $\theta_{wilt}$ is the wilting point soil moisture content below which no root water uptake occurs (-15000 cm in JULES). These reduction factors are multiplied with the root density

in the layer. These weighted reduction factors are then summed to obtain the root zone soil moisture reduction factor. From the actual photosynthesis the plant stomatal conductance is computed. Separately the bare soil surface conductance, which is a function of the soil moisture content in the upper soil layer and the critical soil moisture, is computed. The surface conductance is then computed as a function of the stomatal conductance and the bare soil surface conductance.

The potential evapotranspiration is also calculated separately. This variable is multiplied with the saturated land fraction to

compute the free water evaporation (e.g. lake and canopy evaporation). The rest of the potential evapotranspiration is multiplied with the surface conductance to obtain the bare soil evaporation + plant transpiration. Together these fluxes are the actual evapotranspiration (water) or latent heat flux (energy). The amount of water drawn from the top soil layer through bare soil evaporation depends on the bare soil surface conductance. The distribution of the root water uptake between the layers depends on the weighted soil moisture limitation factor for each layer. The water extraction from the soil in its turn directly

affects the soil moisture content in the different layers at the start of the next time step. These soil moisture contents then affect the soil moisture redistribution, surface runoff, and deep drainage.

JULES-UKV also requires a number of initial conditions: the amount of unfrozen water and snow stored on the surface (on canopy and on soil surface; set to zero in this study), snow properties (set to JULES-UKV defaults), the surface temperature (set to the air temperature of the hour before the first simulation time step), the soil temperature of each layer (set to the soil

temperature from AmeriFlux data the hour before initial time step), the soil water content in each layer (set to the soil water content from the PS observed moisture content of the hour before the first simulation time step and applied homogeneously throughout the profile). Soil moisture was spun up by running a maximum of five cycles and stopped when soil moisture convergence was lower than or equal to 10% compared to the previous cycle.

### 2.4.2 Calibration and validation approaches

At each site we calibrated JULES against PS observed soil moisture and against CRNS observed neutron counts respectively. We chose to calibrate simulated neutron counts against CRNS observed neutron counts using COSMIC (Shuttleworth et al., 2013) to translate simulated soil moisture profiles into equivalent neutron counts. We computed the Root Mean Squared Error (RMSE) between simulated and observed hourly time series. To better match the observed soil moisture/neutron count time series, we calibrated five JULES parameters that influence the model soil moisture state (Figure S1.2 of Supplement 2) These

included Mualem-Van Genuchten shape parameter (b), the water entry pressure parameter (sathh), and the saturation hydraulic conductivity ($K_{sat}$). The critical point ($\theta_{crit}$) and wilting point ($\theta_{wilt}$) soil moisture content parameters from the evapotranspiration

limitation factor were also calibrated. We chose these parameters because they are, in theory, directly linked to the movement of moisture in the soil and to the effects of soil moisture on transpiration in JULES.

To assess the effects of calibration on soil moisture and surface energy flux simulation we compared the calibrated solutions against a default run at each site. The parameter values for the default case were derived from soil properties (percentages clay, loam, and organic matter) reported by the Harmonised World Soil Database (HWSD; FAO, 2009) for each of the twelve sites. These properties were used in the Wösten Pedotransfer Function (Wösten et al., 1997) to obtain values for b, sathh, and $K_{sat}$. Parameter values for $\theta_{crit}$ and $\theta_{wilt}$ were subsequently obtained with the Van Genuchten formula.

The parameter calibration ranges were the same for all sites (Table 2). They were constructed by computing the minimum and maximum parameter values for the entire soil texture triangle (based on Wösten Pedotranser Function). Three organic matter contents were taken into consideration, yielding three triangles. Clay percentages above 70% were excluded to avoid extreme values for parameter b especially. The range for $\theta_{crit}$ was set to 10-90 % of the saturated soil moisture content. The saturated soil moisture content was computed as a function of the dry soil bulk density ($\rho_{bd,dry}$): $\theta_{sat} = 1 - \rho_{bd,dry} / 2.65$ (Brady and Weil, 1996). Soil bulk density values obtained from the COSMOS network were used. This yielded different $\theta_{crit}$ ranges in terms of soil moisture content ($m^3m^{-3}$) for different sites. The residual soil moisture content (defined implicitly in JULES) was set to zero because the Wösten Pedotransfer Function does not consider it.

JULES' remaining two ancillary parameters; the soil heat conductivity and soil heat capacity, were for each site computed as a function of soil properties (HWSD) with De Vries' (1963) method. The bare soil albedo was set constant at 0.38 (-) for all sites. Plant Functional Type parameters were set to JULES defaults except for the e-folding rooting depth (depth above which 86 % of plant roots are present) and the canopy height, at sites where more specific information was available from AmeriFlux/COSMOS site information or from site specific literature.

We calibrated JULES using the BORG Multi-Objective Algorithm (BORG-MOEA or BORG; Hadka and Reed, 2013). This calibration tool was designed for multi-objective problems but also works for single-objective calibration. BORG employs multiple optimisation algorithms simultaneously to obtain convergence while also keeping the searched parameter space wide. The algorithm measures progress with the epsilon-progress technique, which uses the objective function space divided in boxes with sides of size epsilon. Epsilon is a user defined value for each objective function (we used epsilon values of 0.001 $m^3m^{-3}$ for PS and 1 cph for CRNS, Kollat et al., 2012). Only if a new solution resides inside a box with a better objective function value, BORG considers it is progress. If no progress was obtained after 200 runs, the algorithm had stagnated. In this case the BORG algorithm triggers a restart, which consists of (among other techniques) changing population size to maintain a diverse population and to escape local optima. We used a maximum number of 3,000 runs and an initial population size of 100 runs.

As validation metric we chose RMSE between the observed and simulated latent heat flux (LE). Because data quality issues with eddy-covariance data are often observed during night time (Goulden et al., 1996; 2006; 2012; Aubinet et al., 2010), we computed these metrics over day time hours only. We defined day time as downward shortwave radiation > 20 $Wm^{-2}$. To avoid extreme RMSE-EF values, we used hours with both observed and simulated LE values ≥ 1 $Wm^{-2}$ only. Otherwise a few hours

with small LE or H values would have dominated the RMSE values, while this would probably have been due to forcing or eddy-covariance data inconsistencies and would not relate to soil moisture temporal variability.

It is known that single-objective calibration is often insufficient to constrain parameters to simulate different states and fluxes well (Gupta et al., 1999). In addition, Vereecken et al. (2008) argued that calibrating soil hydraulic parameters against soil moisture only does not guarantee better surface energy fluxes. To find out whether it was actually feasible to expect better surface energy flux simulation when soil moisture was improved we performed calibrations where we optimised the model for two objectives simultaneously. We employed the BORG algorithm to simultaneously optimise the RMSE of (day time) latent leat flux and the RMSE of all day soil moisture (using PS soil moisture and CRNS neutron counts separately). We analysed the trade-off between the two objective functions. We computed the compromise solution for each two-objective-calibration. Plotted within the normalised two-objective solution space, the compromise solution is the model run which has the smallest distance to the origin. This means no other solution can be obtained that yields a better approximation for one objective function without deteriorating the other.

## 3 Results and Discussion

### 3.1 Soil moisture data analyses

In Figure 2 comparison between PS and CRNS soil moisture time series shows that the seasonal trends of the two soil moisture observation products were similar. The two soil moisture products however also differed from each other, in distinct ways at different sites. PS soil moisture observations were systematically higher than CRNS soil moisture observations at eight of twelve sites. At three sites (DC, SO, CS) PS soil moisture dried down quicker than CRNS soil moisture, while at ME the opposite behaviour was observed. At KE, MM, TR, and MO PS showed a higher seasonality signal (up to 50% higher) than CRNS. Peaks in PS soil moisture were at three sites (UM, KE, TR) up to twice as high as in CRNS soil moisture. In addition, the CRNS data appears noisier than the PS data, which is an effect of inherent randomness in neutron radiation reaching the CRNS sensor element (Zreda et al., 2012). This effect was more pronounced for lower neutron intensity.

The differences seen in Figure 2 are also summarised in Figure 3, which shows a gradual site to site increase in RMSD between PS and CRNS soil moisture (RMSD-SM$_{obs}$). Overall, bias contributed to 50% or more of the total error at seven out of twelve sites.

Additional analyses indicated that differences between the two soil moisture estimates could not be clearly related to any differences in site physical characteristics other than the mean soil wetness. Dominant vegetation type seemed not to have an effect on the similarity between the two soil moisture data products: both forested and grass sites included those with relatively small RMSD-SM$_{obs}$ and those with relatively high RMSD-SM$_{obs}$. Only bare/shrub covered sites (DC, SR, CS) were all below the sites' average RMSD-SM$_{obs}$ of 0.05 m$^3$m$^{-3}$. These four sites were however also relatively dry. Soil type and soil bulk density were also investigated for correlations with RMSD-SM$_{obs}$, but no trends were discovered. Larger differences between

PS and CRNS soil moisture could be expected at sites with more heterogeneous soil or vegetation. Static satellite photos of the sites from the COSMOS website did not indicate systematically more heterogeneous conditions at the sites where PS and CRNS soil moisture differed more. Site info (e.g. topography, presence of rocks) from COSMOS and AmeriFlux did also not clearly show more horizontally heterogeneous soil properties for those sites. The fact that the soil moisture time series of PS and CRNS differed from each other in various ways could be related to a number of issues. First, PS sensor types and numbers of sensors differed between the sites (Table A1 of Appendix A). Secondly, the exact installation locations of the PS sensors may in certain cases have been for instance next to a macropore, or near roots, while at other sites they were coincidently installed in a homogeneous soil patch.

The presence of neutron mitigating factors other than soil moisture (e.g. atmospheric pressure, sensor type, biomass, intercepted water, and water in litter layer) also affect the observed CRNS neutron count. Because different hydrogen pools are more present at certain sites than at others, the uncertainty on neutron count observations varies between sites. The results did however not show effects of land cover and soil properties on the similarity between the two soil moisture products. Another factor is that, the quality of the calibration of COSMIC could possibly be different for different sites. Finally, at multiple sites, the PS and CRNS soil moisture time series were similar as shown with RMSD values. This could be expected at rather homogeneous sites. Moreover, Köhli et al. (2015) suggested the CRNS footprint to be around 150-200 m instead of 300 m as reported by Desilets and Zreda (2013). In that case the differences between the two soil moisture observation techniques could be smaller than initially thought.

We derived vertically constant CRNS soil moisture values from observed neutron counts with COSMIC. This method contains inherent uncertainty because in reality soil moisture is often not vertically homogeneous. The outcomes of the calibration (against PS and CRNS) and validation provide insight in the effects of the differences between the two soil moisture products on JULES' surface energy flux partitioning and latent heat flux simulation.

### 3.2 Single objective calibration against soil moisture observations

The degree to which the objective function (RMSE-SM; Figure 4) values decreased differed between sites and the two calibration strategies (PS or CRNS), with decreases of 21% (AR-PS) to 93% (UM-CRNS). While the errors of the default runs existed mostly of systematic bias, after calibration the difference in dynamics was the largest source of uncertainty and in 16 out of 24 cases this contribution actually increased in absolute terms. The increase in difference in dynamics was due to the selected objective function (RMSE), which reduces the mean error between modelled and observed data. Previous research (e.g. Teuling et al., 2009) has shown calibrating soil parameters has a large effect on simulated absolute soil moisture values , but substantially less on soil moisture seasonality and dynamics. Our finding supports this.

The RMSE values reduced relatively more for the CRNS calibration (70% on average over the twelve sites) than for the PS calibration (55% on average). The calibration method could possibly explain this. CRNS calibration was against observed neutron counts, while PS calibration was against observed soil moisture contents. Because neutron counts have an inverse relationship with soil moisture content, the PS calibration was possibly governed by avoiding larger errors occurring during a

few brief soil moisture peaks. While focussing on getting the fitting for those peaks right, the PS calibration neglected the smaller errors during dry periods. This would then result in relatively smaller decrease in RMSE values than for the CRNS calibrations because those were fitted with heavier weights to the drier periods.

Figure 4 also shows that the relative improvement was not systematically lower or higher for sites with higher similarity between the two observed soil moisture time series. Actually the largest improvement was for the CRNS calibration at UM. Therefore, it appears that the quality of the default runs was hence predominated by the quality of the chosen default parameter values.

Figure 5 shows soil moisture time series for four selected sites: UM was chosen because PS and CRNS soil moisture were most similar there, SR was a site with moderate difference, and at WR and MO PS and CRNS soil moisture were most different. Simulated soil moisture improved especially at UM and WR, where the default runs overestimated soil moisture contents. Simulated soil moisture dynamics became more similar to both PS and CRNS observed soil moisture dynamics, even though observed soil moisture dynamics differed from each other substantially at sites WR and MO. The relatively high noise in the CRNS soil moisture time series is due to the relatively low neutron count at this site, and possibly due to temporal variations in other hydrogen pools like water intercepted in the forest canopy and in the litter layer.

### 3.3 Validation of the single-objective calibrations against eddy-covariance observations

While calibration errors decreased for soil moisture, latent heat flux estimation improved for fourteen out of twenty-four calibrations (Figure 6). This means that an improvement in simulated soil moisture did not necessarily lead to better estimation of surface energy fluxes. Calibration against PS soil moisture improved RMSE-LE at six sites (UM, KE, DC, SR, SO, and ME), while calibration against CRNS neutron counts improved RMSE-LE at eight sites (KE, DC, SR, ME, SO, AR, MM, and WR). At five sites RMSE-LE improved after calibration against both PS soil moisture and CRNS neutron counts. Calibration yielded lower RMSE-LE after calibration against CRNS neutron counts than after calibration against PS soil moisture at all but three sites (UM, KE, and WR). Figure 6 also shows that RMSE-SM decreased substantially less (i.e. >20% difference) after calibration against PS soil moisture than after calibration against CRNS neutron counts (i.e. >20% difference between both strategies on the horizontal axis of Figure 6, which occurred at six sites). At five of these six sites the relative change in surface energy flux performance was also smaller (sites MO, AR, TR, ME, and SR). This indicates that further improvement in soil moisture simulation after calibration against PS data could have yielded better surface energy fluxes. The differences between the two calibration strategies can also be seen from the different locations of the centres-of-mass of the two point clouds. The cloud of the CRNS strategy is clearly located more to the lower left corner in Figure 6.

In Figure 6 10% change in latent heat flux estimation was chosen to distinguish substantial change from non-substantial change, derived from the approximate error in eddy-covariance data (Sellers and Hall, 1992; Finkelstein and Sims, 2001). Improvement in latent heat flux was actually substantial in four cases for PS calibration and five cases for CRNS calibration. Using this threshold also revealed that RMSE-LE did not change substantially in six cases for calibrations with a more than 60% change in RMSE-SM. This again shows that a change in simulated soil moisture did not necessarily mean a substantial change in

surface energy flux simulation. Analysis of the RMSE of evaporative fraction (EF=LE/(LE+H)), which shows the ability to simulate surface energy partitioning, yielded similar overall results as our analysis of the RMSE of latent heat flux (Supplement 2, Figure S2.1).

One factor causing some of these limited improvements was that, when mean simulated root zone soil moisture (weighted with

root density) increased after calibration, the values of the wilting point and critical point soil moisture parameters moved along (data not shown), yielding similar soil moisture stress. This happened for both calibration strategies at site KE, and for the calibration against CRNS neutron counts at sites MO and TR. This could relate to the limited value of simulated absolute soil moisture for surface energy flux estimation in land surface models (Dirmeyer et al., 2000; Koster et al., 2009). However, we also found the distance between wilting point values and critical point values to decrease after calibration. This occurred with

a simultaneous decrease in standard deviation of the simulated soil moisture. The self-adjusting behaviour of the wilting point and critical point parameters was also indicated by parameter sensitivity analysis (Appendix 2), which showed soil moisture was substantially more sensitive to a change in critical point value than latent heat flux.

Another issue, which occurred for instance for the PS calibration at SO and for the CRNS calibrations at site AR, was that while surface energy flux estimation improved for a certain period, it deteriorated for another period (data not shown). A third

cause of limited improvement in surface energy flux estimation occurred for example at site ME (data not shown). Soil moisture stress was relatively limited (beta mostly above 0.6) and during periods of soil moisture stress, the latent heat flux was dominated by different stress factors.

To see if validation results would be different if only those periods during which soil moisture stress occurred were evaluated,

we also analysed the performance over these periods (data not shown). During these periods plant water uptake limitation factor β was below one (data not shown). Only at site SO did we see somewhat better performance during these periods compared to the original validation period, after both PS and CRNS calibration.

We explored trends between relative improvement in surface energy flux estimation and soil wetness, precipitation, vegetation type, vegetation height, and soil characteristics. No clear trends were discovered. The only feature that could be distinguished

was that for the two sites with a bare soil tile (DC and SR) PS and CRNS calibration improved EF and LE. Rooting depth did also not explain relative improvement in surface energy flux estimation. Finally, larger differences between the two observed soil moisture time series did generally yield more different simulated surface energy flux time series, with a clear exception for site UM, where the two calibration approaches yielded more than 20% difference in change in RMSE-LE.

In Figure 7 the monthly mean diurnal latent heat cycles of four sites (UM, SR, WR, and MO) are shown (night time data

included but not used for calibration). Both calibration strategies yielded overestimation in March and April and underestimation from June to August at site UM. A month long gap in longwave and shortwave downward model forcing data occurred in April 2012. This might have affected the results for this site. At SR, both PS and CRNS calibration improved latent heat flux during periods of low evapotranspiration, while during the other periods only the CRNS calibrations yielded better results. At WR calibration against PS soil moisture yielded overestimation of latent heat flux, while CRNS calibration yielded

too low latent heat flux. The PS calibration at MO yielded latent heat flux underestimation, whereas CRNS calibration did not change LE substantially.

In summary, the single-objective calibrations against CRNS neutron counts yielded larger improvements in simulated surface energy fluxes than single-objective calibrations against PS soil moisture (Figure 6). Improvements in surface energy flux

estimation were however substantial for four calibrations against PS soil moisture and five calibrations against CRNS neutron counts. Limited improvements in surface energy flux estimation after calibration could partly be attributed to the limited value of absolute soil moisture to estimate surface energy fluxes with land surface models. This seems reasonable because calibration mostly affected absolute soil moisture (Figure 4). This result corresponds with earlier research that showed model soil moisture dynamics and seasonality have a larger effect on surface energy flux simulation than absolute soil moisture (e.g. Teuling et al.,

2009; Dirmeyer et al., 1999). Previous research has also indicated that soil moisture alone is insufficient to estimate soil hydraulic parameters (Vereecken et al., 2008; 2015). Our findings support this for some sites. To better understand these implications, the two-objective simulations against soil moisture and latent heat flux simultaneously (discussed in the next section) provides further insight in these results.

### 3.4 Two-objective calibration against soil moisture and latent heat flux

The results from the two-objective calibrations suggest that latent heat flux estimation improvement with respect to the compromise solutions (%-change RMSE-LE, vertical axis in Figure 8) was similar for both two-objective calibration strategies (Figure 8). Only at three sites the difference in improvement (%-change RMSE-LE, vertical axis in Figure 8) between the two-objective calibration strategies was more than 5% (at sites SR, DC, and UM). Improvements were substantial (according to our 10% threshold for improvement in RMSE of latent heat flux) for six calibrations with PS soil moisture and for eight

calibrations with CRNS soil moisture.

Scatterplots of normalised RMSE of latent heat flux (LE) and normalised RMSE of soil moisture for all sites are shown in Figure 9 and Figure 10 for the PS and CRNS two-objective calibration strategies respectively. The RMSE values were normalised with respect to the default solutions, which therefore have normalised RMSEs of 1 (-). The black dots represent individual model runs and the default model run for each site is represented by a red cross. The single-objective calibration

solution of each single-objective calibration solution is shown as a blue triangle and the compromise solution of each two-objective calibration is shown as a green triangle. Figure 9 and Figure 10 show that the substantial (i.e. more than 5%) differences between the two two-objective calibration strategies in improvement of simulated latent heat flux observed for SR, DC, and UM in Figure 8, were less meaningful than seemed initially from analysis of Figure 8. These findings are based on the shapes of the black point clouds in Figure 9 and Figure 10. We first look at the left edges of the black point clouds. When

these edges are close to vertical, a small deterioration in RMSE-SM or RMSE-N (e.g. less than 0.05 (-)) would yield a large deterioration in normalised RMSE-LE (e.g. greater than 0.2 (-)). We observed this for three calibrations with PS soil moisture (SO, AR, and WR; indicated with pink line for WR) and in all these three cases, a less than 0.05 (-) change in normalised RMSE-SM would have yielded worse simulated latent heat flux than with the default parameter set. Seven calibrations with

CRNS neutron counts (UM, DC, SO, KE, SR, CS, and WR; indicated with pink line for WR) showed a close to vertical edge. In four cases (UM, DC, KE, and CS) this would have yielded worse simulated latent heat flux than with the default parameter set. A negative slope for this side of the point cloud means an improvement in soil moisture estimation would mean a deterioration in latent heat flux estimation. We observed such negative slope for four calibrations with PS soil moisture (CS,

MM, TR, and MO; indicated with continuous black line for MO) and for two calibrations with CRNS neutron counts.

Next, we look at the lower edges of the point clouds in Figure 9 and Figure 10. When this edge is horizontal, this means good latent heat flux estimation can be obtained for a wide range of soil moisture estimation performances (for instance site ME, indicated with pink lines in both figures). When this edge has a negative slope, this means the best latent heat flux estimation is obtained for worse soil moisture (for instance site TR, indicated with pink lines in both figures). We observed these two

features for all two-objective calibrations. Single-objective calibration against latent heat flux would however only have necessarily yielded worse soil moisture than the default parameter set for six calibrations with respect to PS soil moisture (sites UM, KE, SR, TR, AR and WR) and two calibrations with respect to CRNS neutron counts (sites TR and AR). The implication of these results is that the quality of latent heat flux simulation did not depend strongly on the quality of soil moisture simulation.

In summary, the two-objective calibrations against soil moisture (or neutron counts) together with latent heat flux showed, compared to the single-objective calibrations, fewer substantial differences between calibration with PS soil moisture and calibration with CRNS neutron counts. These results indicate that the differences between both *single*-objective calibration strategies, which showed an advantage for CRNS observations, were possibly not as substantial as it seemed at first. The spatio-temporal stability theory, which implies limited spatial variability in surface energy fluxes, could be one explanation

for this (Vachaud et al., 1985; Teuling et al., 2006; Mittelbach and Seneviratne, 2012; Albertson and Montaldo, 2003). Another factor that has possibly played a role is that the spatial scale advantage of the CRNS was masked out by the possibly lower quality of its measurements (see also Section 3.1). Different hydrogen pools than soil moisture affect the neutron measurements at various temporal resolutions (e.g. rainfall interception; Baroni and Oswald, 2015). However, PS sensors also have their limitations. Different (electromagnetic) PS sensors have different designs and properties, which affect the data quality

(Robinson et al., 2008; Blonquist et al., 2005). Soil type and soil specific calibration also affect the accuracy and precision of the PS data. For instance, the relationship between electrical permittivity and soil moisture content is strong for quartz rich soils, but less accurate for clay soils (e.g. Ishida et al., 2000; Robinson et al., 2008). An issue that could have had effect on our results is the occurrence of gaps in the model forcing data. Gaps in observed meteorological data are often inevitable and must be filled for a land surface model to be run. Percentages of missing hours differed between 0% and 15% at our sites. Gaps

larger than fifteen days filled with the moving window gap-filling procedure occurred mainly for downward shortwave and/or downward longwave radiation; at sites UM, KE, SR, DC, TR, and CS. At site WR a data gap of 29 days in precipitation and air pressure occurred. Especially the gap in precipitation data can have negatively affected the model results at this site. Gaps larger than 30 days were filled with NLDAS-2 data at sites WR, CS, and SO. At site WR a single gap of 110 days in shortwave and longwave radiation was filled. At site SO a gap of 66 days in atmospheric pressure was filled with NLDAS-2 data. At site

CS two gaps of 200 days in atmospheric pressure was filled and the entire time series of air temperature was filled with a linear relationship based on data from four preceding years. At site SR, gaps before applying the moving window procedure varied between 3% and 15 % of the time for precipitation and wind speed, respectively and atmospheric humidity was missing for 35% of the time. These gaps were mostly filled with a linear relationship with data from the Sahuarita site, located approximately 14 km to the north-west from the SR site, except for downward longwave radiation. Remaining gaps were filled with the moving window procedure. In this study we used PS sensor data in the upper 30 cm of the soil only, even though data from deeper sensors was publicly available at two sites (WR and MO). This choice can have provided a disadvantage to the PS data in our comparison, especially at sites with deeper roots and during soil moisture limiting conditions in the upper soil. Investigating the role of deeper roots was however beyond the scope of this study. Our goal was to compare the effects of the difference in horizontal footprint on latent heat flux simulation after using the two different measurement techniques to calibrate model parameters.

## 4 Conclusions

We investigated whether the spatial scale mismatch between the surface energy flux data and soil moisture data could be reduced through the use of Cosmic-Ray Neutron Sensors (CRNS). Five soil and evapotranspiration parameters of LSM JULES were calibrated against Point Scale (PS) and CRNS soil moisture data separately, for twelve COSMOS/AmeriFlux sites with different climate, land cover, and soil properties. Next, at each site, the improvement in latent heat flux for the two calibration solutions was assessed by comparing the fit with eddy-covariance data and a version of LSM JULES runs with default parameter values based on a widely used soil database. Before the calibrations we compared the observed soil moisture data from the two sensor types. These analyses showed the differences between PS and CRNS soil moisture varied in nature at the investigated sites but such differences were however small. While at eight sites there were mainly systematic biases, at three other sites the seasonality was most different, and at one other sites different time series dynamics were the main cause of differences between the two soil moisture observations. The single-objective calibration of JULES parameters against Cosmic-Ray Neutron Sensor neutron counts yielded better simulated latent heat flux than single-objective calibration against point-scale soil moisture. The analysis of multi-objective calibrations (against (1) PS soil moisture and latent heat flux and (2) CRNS neutron counts and latent heat flux) however revealed that differences between calibrations with these two soil moisture observation methods did overall not yield substantially different surface energy flux estimations.

These outcomes did not provide sufficient evidence to reject our null-hypothesis "*reduced scale mismatch does not lead to LSM flux estimates closer to eddy covariance observations*". The spatio-temporal stability theory in soil moisture, on which our hypothesis was based, can possibly explain the limited differences in surface energy flux estimation. This theory implies that spatial variability in surface energy fluxes is relatively limited within an eddy-covariance tower footprint. Therefore simulated surface energy fluxes after calibration with point scale soil moisture data would not necessarily be better than simulated surface energy fluxes after calibrations with CRNS neutron count data. Calibrating soil parameters had mostly an

effect on absolute soil moisture values rather than soil moisture dynamics. Soil moisture dynamics have a greater effect on surface energy flux simulation in land surface models than absolute soil moisture values do. Related to this we observed that after calibration the wilting point and critical point soil moisture parameter values adjusted themselves in a similar way as the root zone soil moisture did, yielding similar soil moisture control on transpiration despite changes in soil moisture values. In other cases we found calibration against soil moisture to improve surface energy fluxes during certain periods, but to deteriorate surface energy fluxes during other periods, yielding similar overall performance. Yet in other cases evapotranspiration was not limited by soil moisture stress. The potential scale advantage of the CRNS was possibly masked out by the possibly lower measurement quality of this sensor because other hydrogen pools than soil moisture affect the neutron count observations. Future use of CRNS soil moisture data could however benefit from improved knowledge on the effects of additional hydrogen pools (e.g. Baroni and Oswald, 2015) and of the sensor footprint (Köhli et al., 2015). In this study, our results are conditioned to a single land surface model (JULES). For additional understanding of the importance of both PS and CRNS measurements for surface energy flux simulation, this study can be extended to other models in the future.

**Acknowledgements**

This research was supported by the Queen's School of Engineering (University of Bristol) Ph.D. scholarship and the by Engineering and Physical Sciences Research Council (EP/L504919/1). Additional support for this work was also provided by the Natural Environment Research Council (A MUlti-scale Soil moisture-Evapotranspiration Dynamics study (AMUSED); grant number NE/M003086/1). Funding for AmeriFlux data resources was provided by the US. Department of Energy's Office of Science. We would like to thank the investigators of the sites used for providing us with site information through personal communication. The authors would like to thank Jos van Dam, referee Ryan Teuling, referee Roland Baatz, and one anonymous referee for providing constructive comments. Finally, the authors would like to thank the Editor, Harrie-Jan Hendricks Franssen, for guiding the revision process.

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

**Table 1: Site characteristics. Altitude from COSMOS website, land cover percentages from AmeriFlux and publications. Harmonised World Soil Database (HWSD) data was used to define default model parameter values (here only soil categories shown).**

| Site | Altitude (meter above sea level; COSMOS) | Land cover (%) | | HWSD dominant soil type | Site soil info (AmeriFlux, COSMOS, literature) | Data sources |
|------|------|------|------|------|------|------|
| | | Dominant | Remaining | | | |
| UM | 220 | 100% broadleaf | | Loamy sand | Deep well drained soils | COSMOS, ORNL-DAAC (2015) |
| DC | 1300 | 46% shrubs | 46% bare, 8% needleleaf | Sandy loam | | Anderson and Goulden (2012) |
| SO | 1160 | 63% needleleaf | 37% shrubs | Loam | 0-10 cm thick organic litter layer, bedrock at 1-2 m | COSMOS, Goulden et al. (2012) |
| KE | 1531 | 100% C4-grass | | Loam | Coarse-loamy, limestone fragments | ORNL-DAAC (2015) |
| ME | 1253 | 100% needleleaf | | Loamy sand | Sandy, minimal organic | COSMOS, ORNL-DAAC (2015) |
| SR | 989 | 76% bare | 24% shrubs | Loam | Silty clay loam | COSMOS, Cavanaugh et al. (2011) |
| CS | 457 | 100% shrubs | | Loam | | Anderson and Goulden (2012) |
| MM | 288 | 100% broadleaf | | Loam | Well drained clay loam | COSMOS, ORNL-DAAC (2015) |
| TR | 177 | 60% C3-grass | 40% broadleaf | Loam | Sandy clay loam, clay hardpan at 30-40 cm | COSMOS, ORNL-DAAC (2015), Chen et al. (2008) |
| AR | 314 | 100% C4-grass | | Loam | Sandy | COSMOS, ORNL-DAAC (2015) |
| WR | 371 | 100% needleleaf trees | | Loam | 5-10 cm organic layer, silty sand | COSMOS, ORNL- |

| | | | | | DAAC (2015) |
|---|---|---|---|---|---|
| **MO** | 219 | 100% broadleaf trees | Loam | Silty loam | COSMOS, ORNL-DAAC (2015) |

**Table 2: Calibrated parameter definitions and calibration ranges.**

| JULES parameter name | Unit | Role | Range | |
|---|---|---|---|---|
| | | | Minimum | Maximum |
| **b** | - | Mualem-Van Genuchten parameter (b=1/(n-1)) | 0.63 | 24.43 |
| **sathh** | m | Mualem-Van Genuchten parameter (sathh=$\alpha^{-1}$) | 0.09 | 28.01 |
| **$K_{sat}$** | mm s$^{-1}$ | Mualem saturation hydraulic conductivity | $3 \cdot 10^{-5}$ | $4.3 \cdot 10^{-1}$ |
| **$\theta_{crit}$** | m$^3$m$^{-3}$ | Critical soil moisture content | $0.1 \cdot \theta_{sat}$ | $0.9 \cdot \theta_{sat}$ |
| **$\theta_{wilt}$** | m$^3$m$^{-3}$ | Wilting soil moisture content | $0.1 \cdot \theta_{crit}$ | $0.9 \cdot \theta_{crit}$ |

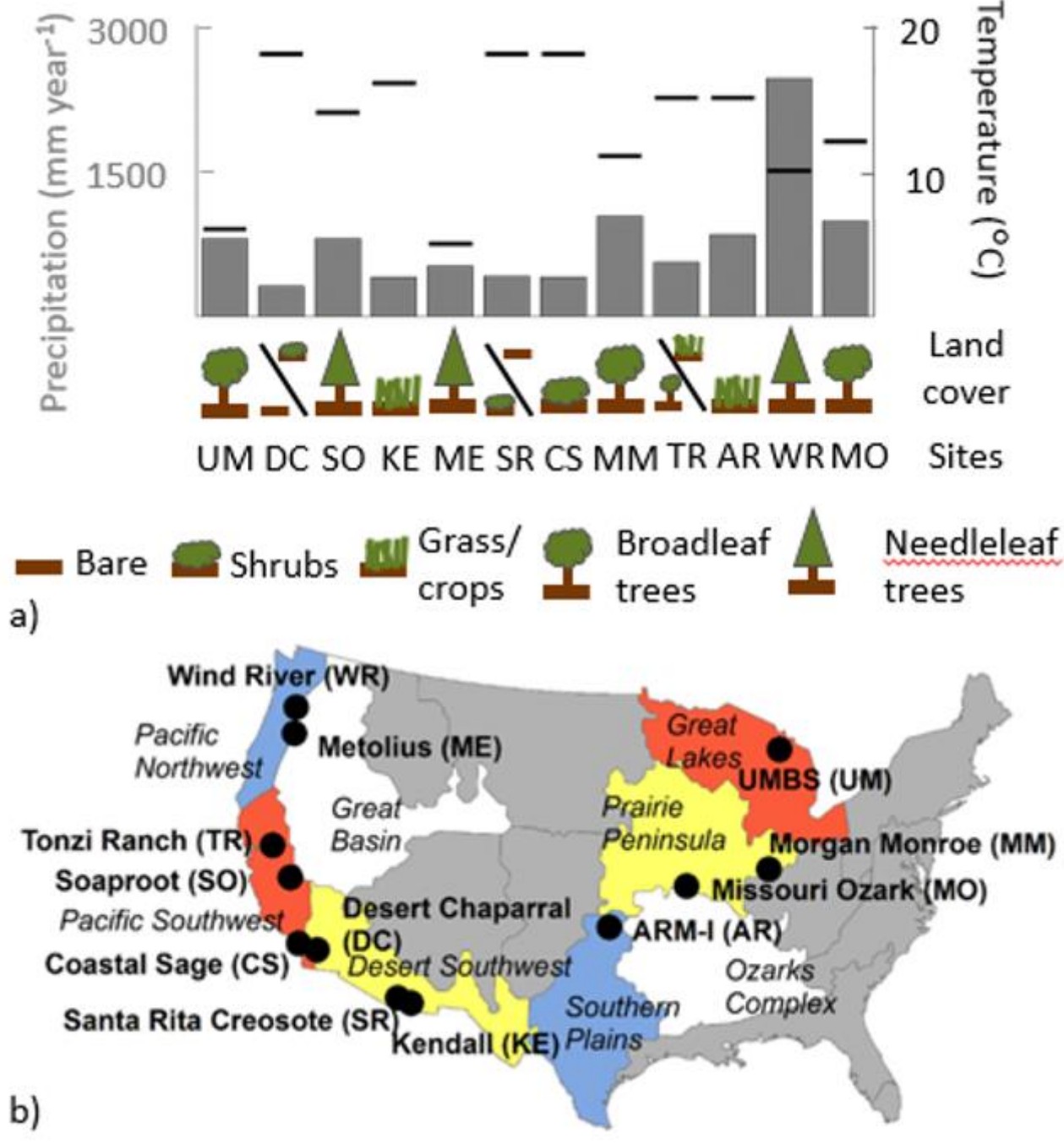

**Figure 1: Figure a shows the yearly mean precipitation, air temperature, and dominant land cover types for the twelve AmeriFlux/COSMOS sites used. At sites DC, SR, and TR two different land cover types were shown because they covered similar areas in size. The map below (b) shows the locations of the twelve sites within eight NEON Ecoclimatic Domains. Data sources: COSMOS, ORNL-DAAC (2015), Goulden et al. (2012), Anderson and Goulden (2012), Scott et al. (1990), Chen et al. (2008).**

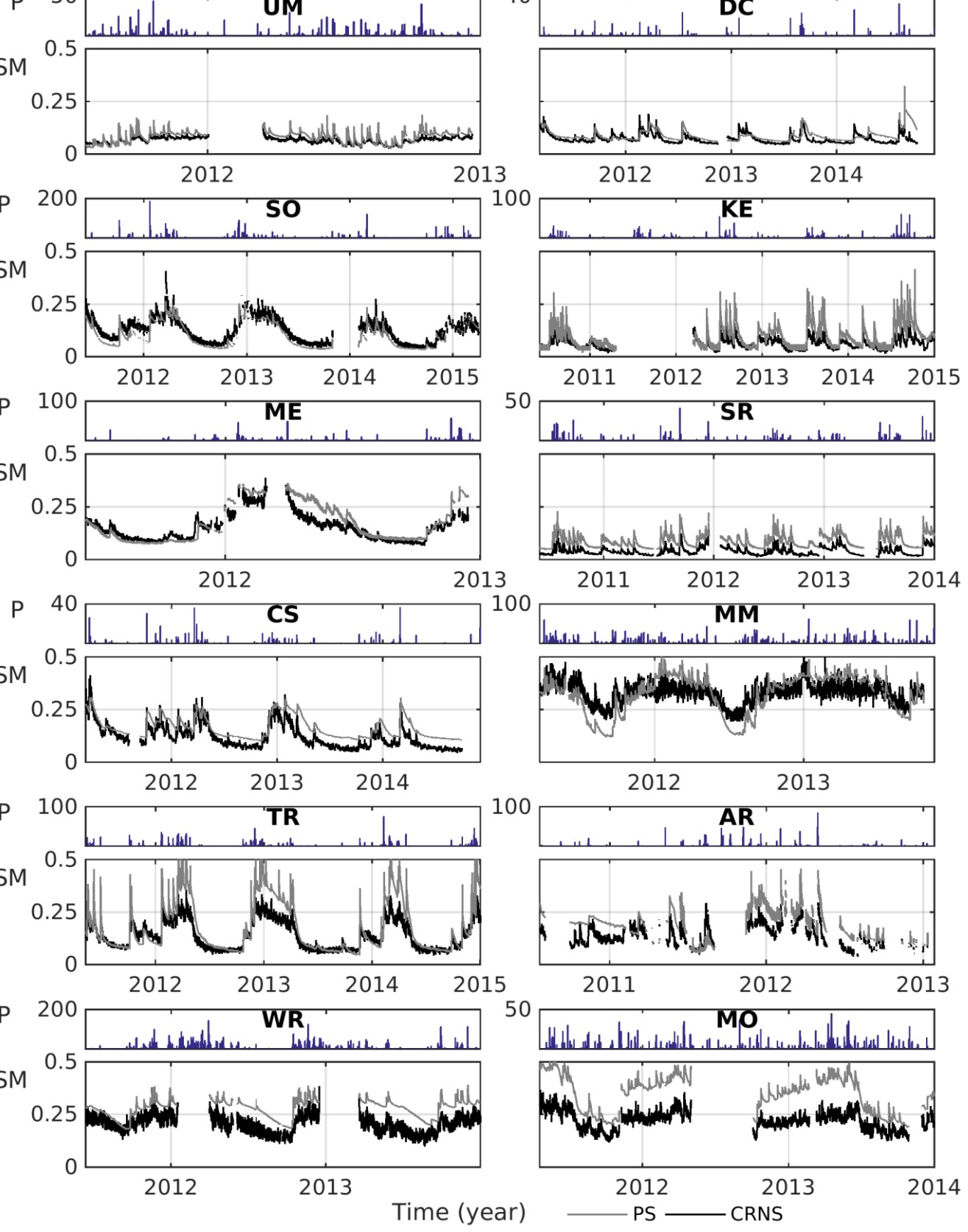

**Figure 2: PS and CRNS observed soil moisture (SM; $m^3m^{-3}$) time series for the twelve study sites. Notice the PS soil moisture time series have been linearly interpolated from individual measurement depths to the corresponding JULES soil layers. CRNS soil moisture was obtained using COSMIC while assuming vertically homogeneous soil moisture. Daily precipitation (mm day$^{-1}$) is also shown here for each site.**

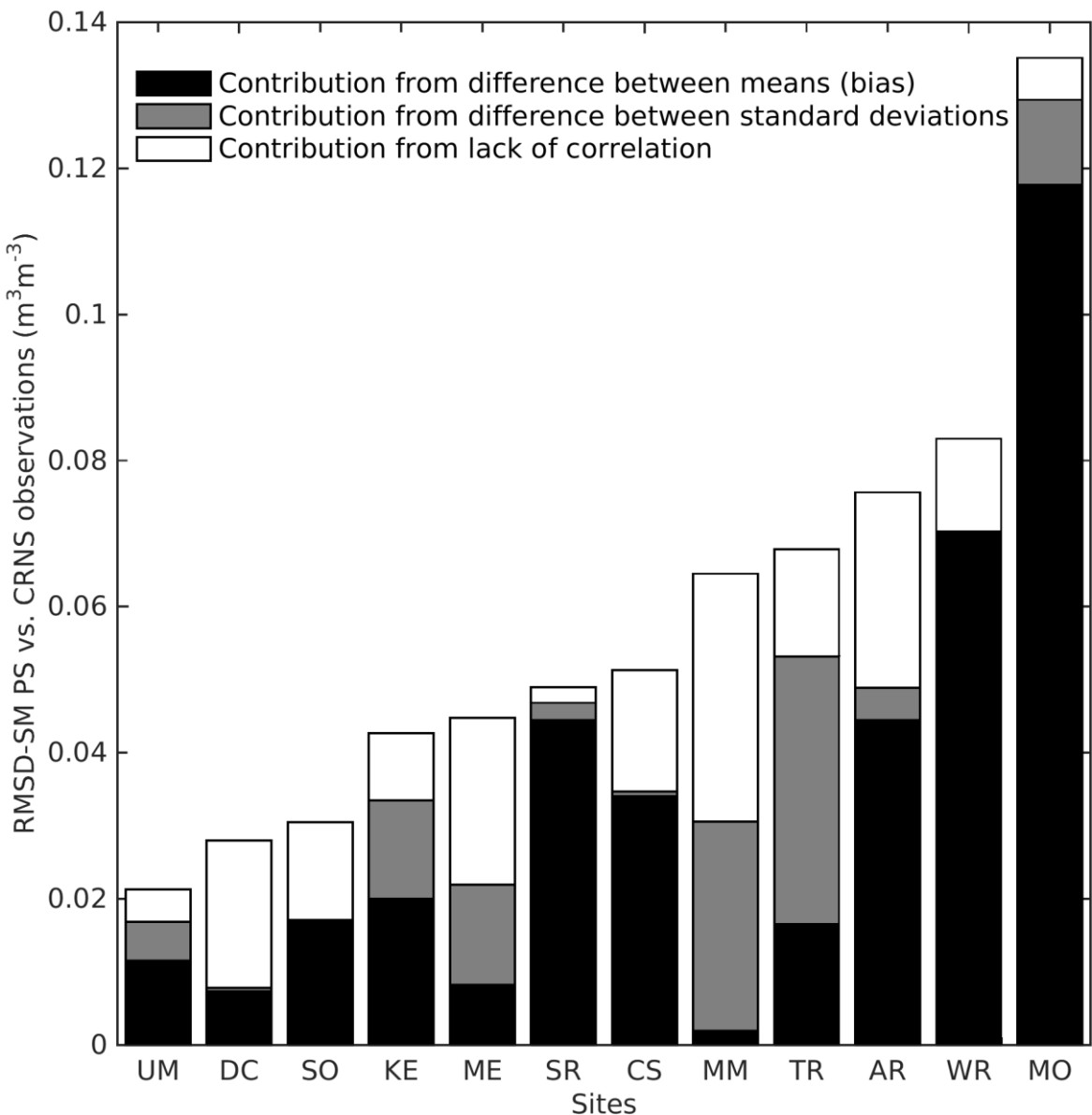

**Figure 3: Root Mean Squared Deviation (RMSD) between observed PS and CRNS soil moisture (SM; m³m⁻³). MSD decomposition (Gupta et al., 2009) was calculated and the fractions were then applied to the RMSD values. Sites ranked from most similar to most different.**

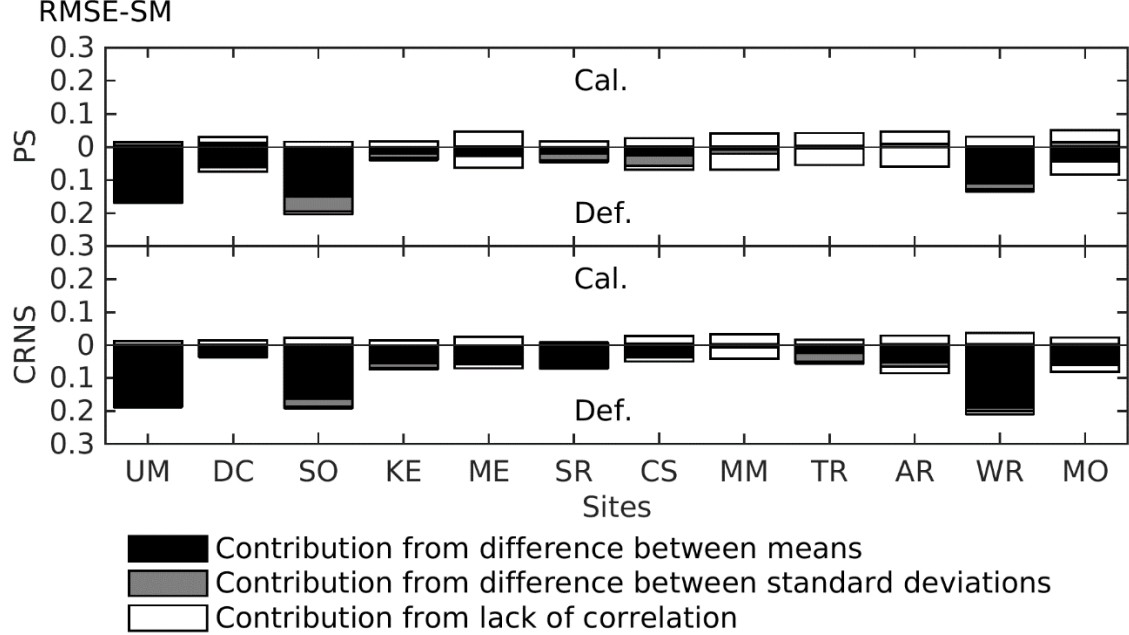

**Figure 4: Objective function (Root Mean Square Error; RMSE (m³m⁻³)) values between observed PS/CRNS soil moisture and JULES simulated soil moisture. For each calibration the RMSE of the default run is shown from the horizontal axis down, and the result after calibration is shown in the upward directions. The different error contributions from the MSE decomposition are shown as stacked bar plots.**

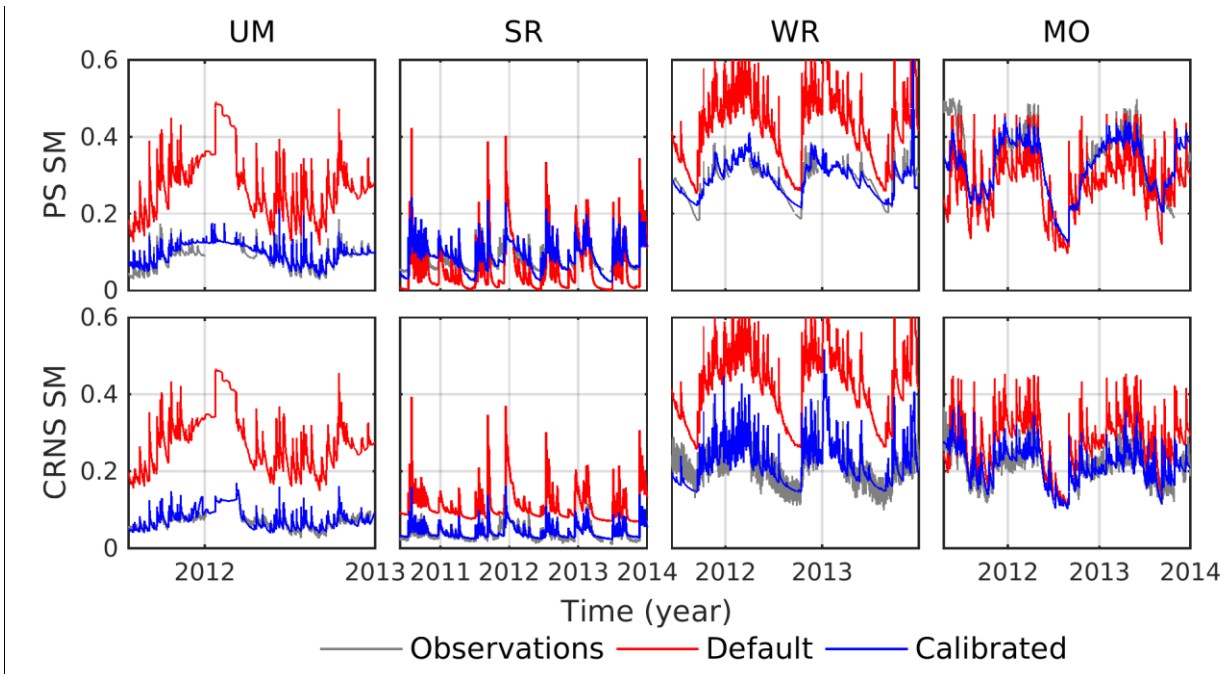

**Figure 5: Hourly soil moisture time series of JULES default and calibrated runs against observations (PS or CRNS) for four of twelve sites.**

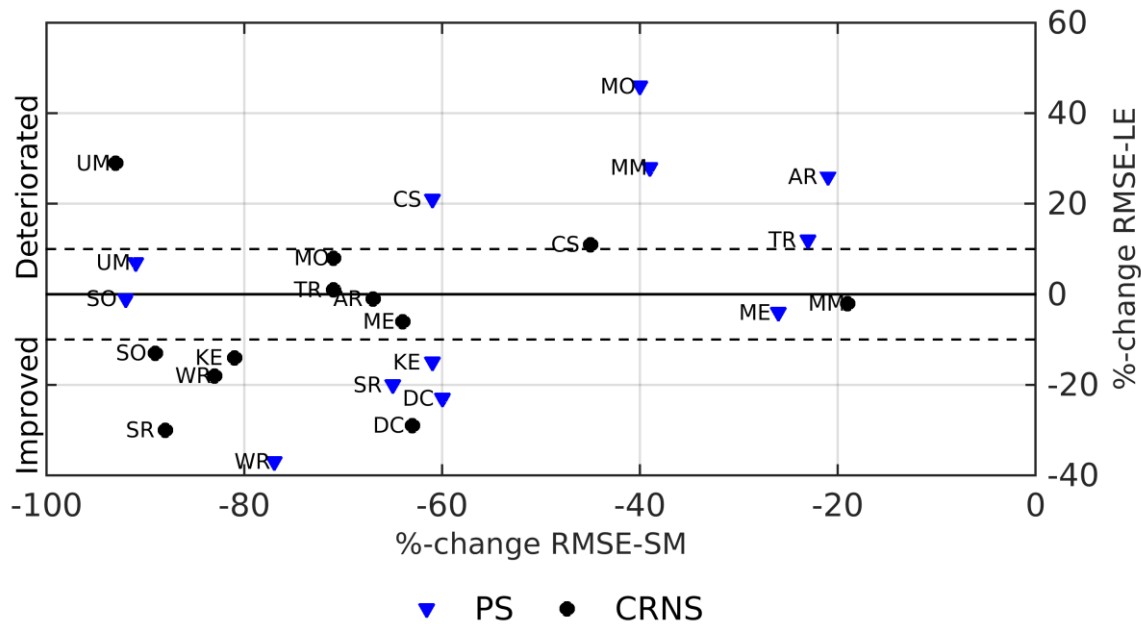

**Figure 6: Relative change in Root Mean Square Error (RMSE) values after single-objective calibration against PS soil moisture and CRNS neutron count data. Change in RMSE between observed and simulated latent heat flux (LE) is plotted against change in RMSE between observed and simulated soil moisture.**

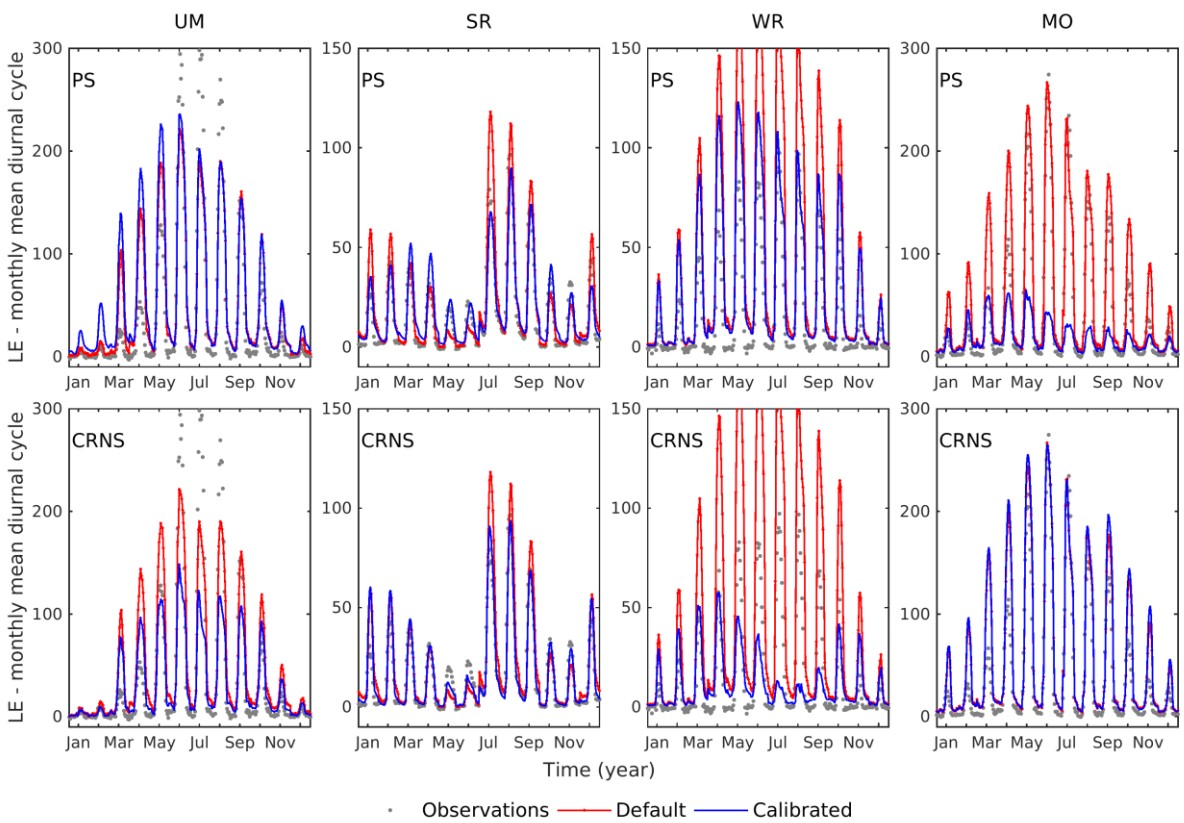

**Figure 7: Monthly mean diurnal latent heat flux (LE) cycles. The upper row contains PS calibrated solutions, the lower row shows the CRNS calibrated solutions. The upper row contains PS calibrated solutions, the lower row shows the CRNS calibrated solutions.**

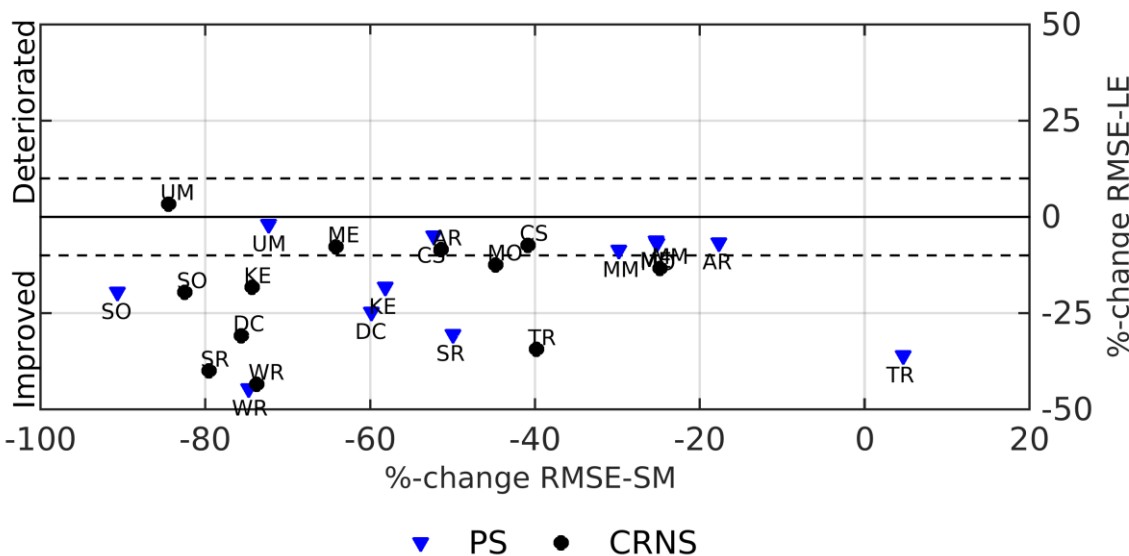

**Figure 8: Relative change in Root Mean Square Error (RMSE) values after two-objective calibration against (1) PS soil moisture and latent heat flux and (2) against CRNS neutron count data and latent heat flux. Change in RMSE between observed and simulated latent heat flux (LE) is plotted against change in RMSE between observed and simulated soil moisture.**

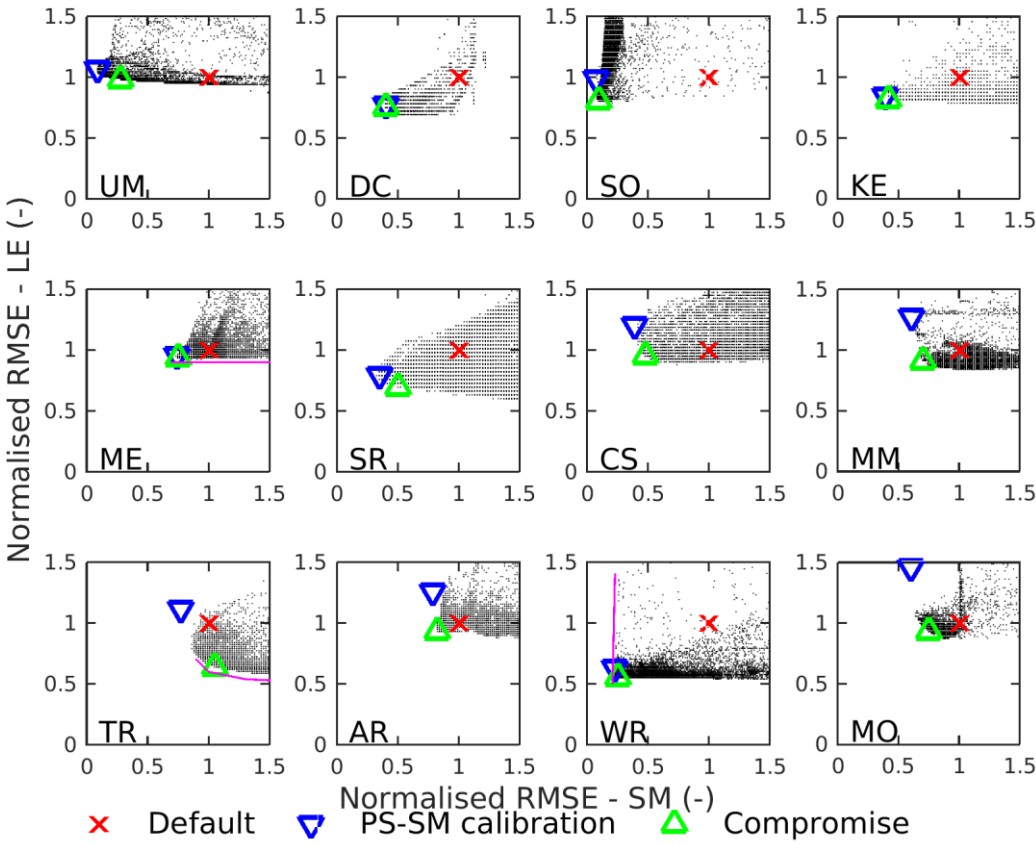

**Figure 9: All runs from the two-objective calibrations against PS soil moisture and day time hourly latent heat flux (LE) at each site. Default run, PS-SM single objective calibrated run, and compromise solution runs are shown on top. All values were normalised to the values of the Default run. The plots were zoomed in to the area of 0 to 1.5 times the Default values.**

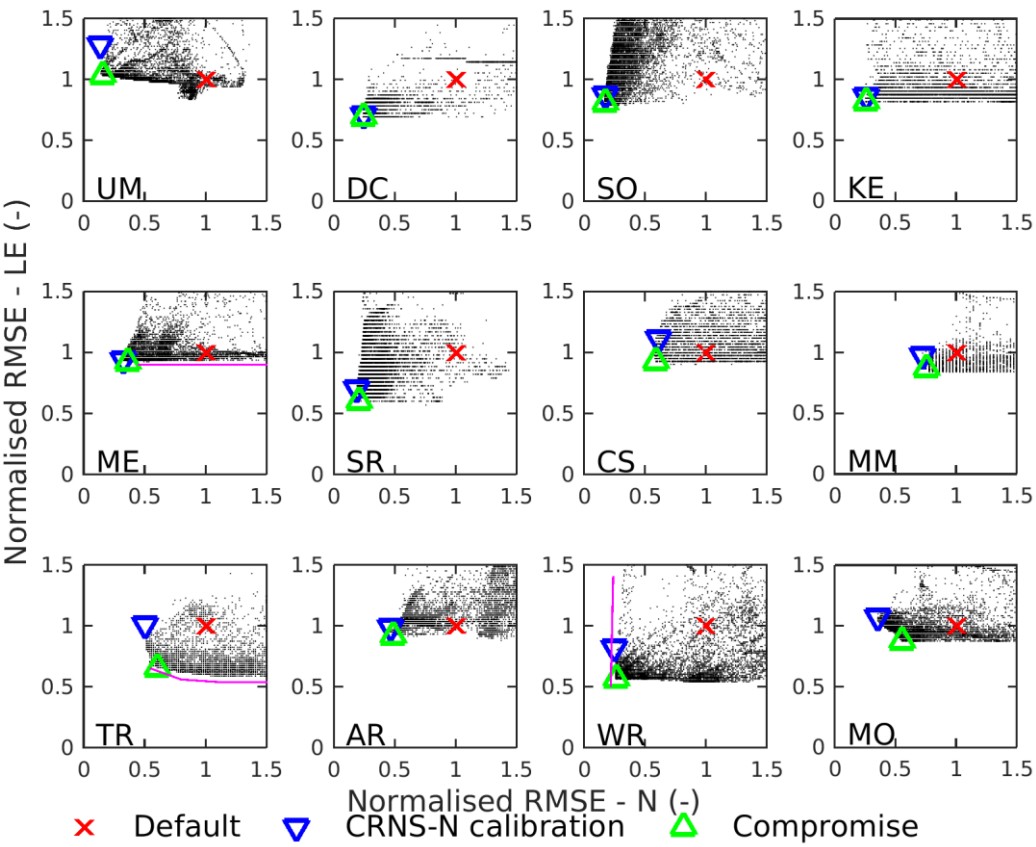

**Figure 10: Same as Figure 9 but for the two-objective calibrations against CRNS neutron counts (representing soil moisture) and day time hourly latent heat flux (LE) at each site.**

# Appendices

## Appendix 1: Additional Data and Methods tables and figures

**Table A1.1: PS types, installation depths, number of profiles. In the last column it is shown to which JULES layer the observations were linearly interpolated.**

| Site | Type | Installation layers (cm below surface) | Number of profiles | Interpolated to JULES layer (cm) |
|---|---|---|---|---|
| **UMBS** | Campbell Scientific CS615 / CS616 (reflectometer) | 0-30 average | 1 | 0-10 |
| **Desert Chaparral UCI** | Campbell Scientific CS616 (reflectometer) | 0-30 average | 4 | 0-10 |
| **Soaproot** | Campbell Scientific CS615 (reflectometer) | 0-30 average | 4 | 0-10 |
| **Kendall** | Stevens Water Hydra Probe | 5,15 | 1 | 0-10 |
| **Metolius** | Campbell Scientific CS615 (reflectometer) | 0-30 average | 1 | 0-10 |
| **Santa Rita Creosote** | Time Domain Resistivity | 2.5,12.5 | 6 | 0-10 |
| **Coastal Sage UCI** | Campbell Scientific CS616 (reflectometer) | 0-30 average | 4 | 0-10 |
| **Morgan Monroe** | Campbell Scientific CS615 (reflectometer) | 0-30 average | 1 | 0-10 |
| **Tonzi Range** | ThetaProbe (ML2) | 0,(20) | 2 | 0-10,(10-35) |
| **ARM-1** | Decagon Echo2 EC-20 (capacitance) | 10,25 | 2 | 10-35 |
| **Wind River** | Campbell Scientific CS615 (reflectometer) | 0-30 average | 1 | 0-10 |
| **Mozark** | Delta-T | 10 | 1 | 0-10 |

**Table A1.2: Data used per site for downward longwave radiation (L$_{w,in}$), net radiation (R$_{net}$), and atmospheric pressure (P$_{atm}$). A is AmeriFlux Level 2, COSMOS is COSMOS website, and NLDAS is National Land Data Assimilation Systems Forcing data Phase 2 (http://disc.sci.gsfc.nasa.gov/uui/datasets?keywords=NLDAS).**

| Site | L$_{w,in}$ | R$_{net}$ | P$_{atm}$ |
|------|-----------|-----------|-----------|
| UM | A | | A |
| DC | | A | COSMOS |
| SO | NLDAS | | A |
| KE | | A | A |
| ME | A | | A |
| SR | A | | A |
| CS | NLDAS | | COSMOS |
| MM | A | | A |
| TR | NLDAS | | A |
| AR | A | | A |
| WR | A | | A |
| MO | A | | A |

## Appendix 2: Parameter sensitivity analysis

We investigated if a change in some parameters had a relatively small effect on soil moisture, while at the same time having a large effect on latent heat flux. In such case calibration could yield better soil moisture, but the parameter value might be inappropriate for latent heat flux. The inverse (influential on SM but not on LE) could also occur. We explored this by performing a sensitivity analysis with Morris' method (Morris, 1991) as implemented in the SAFE Toolbox (Pianosi et al., 2015) on the exact same parameter value ranges as used during the calibration. We computed the sensitivity indices (mean and

standard deviations of the elementary effects) on the RMSE values (i.e. our Objective Functions) of simulated vs. observed PS soil moisture, simulated vs. observed CRNS neutron counts, and simulated vs. observed latent heat flux (day time only).

The results (shown for four sites in Figure A2.2 of Appendix 2) were consistent across most sites: all three Objective Functions were most sensitive to changes in parameter b and least sensitive to the wilting point multiplier. Finch and Haria (2006) also found JULES parameter b to be most influential on soil moisture and latent heat, at a UK chalk site. The critical point soil

moisture content was influential with respect to soil moisture / neutron counts but had at all sites less effect on latent heat flux. This could relate to the self-adjusting behaviour of the wilting point and critical point soil moisture described in Section 3.3. The lack of effect from the wilting point soil moisture content can probably be attributed to the use of the multiplier, despite being a common approach (Prihodko et al., 2008; Rosolem et al., 2012); a certain multiplier value could be good in combination with a certain value for the critical point but not for a different critical point value.

We have evaluated the differences between default and calibrated parameters and their degree of physical realism in Supplement 3. If parameter values obtained after calibration were not physically feasible (e.g. representing a sandy soil while there was a clay soil) then, if model structure is assumed to represent biophysical processes sufficiently well, that could yield undesirable results. The wide ranges of parameter values, especially for the water entry pressure (sathh) and the saturation

hydraulic conductivity. However, physically realistic parameters also often provide unrealistic results (Gupta et al., 1998).

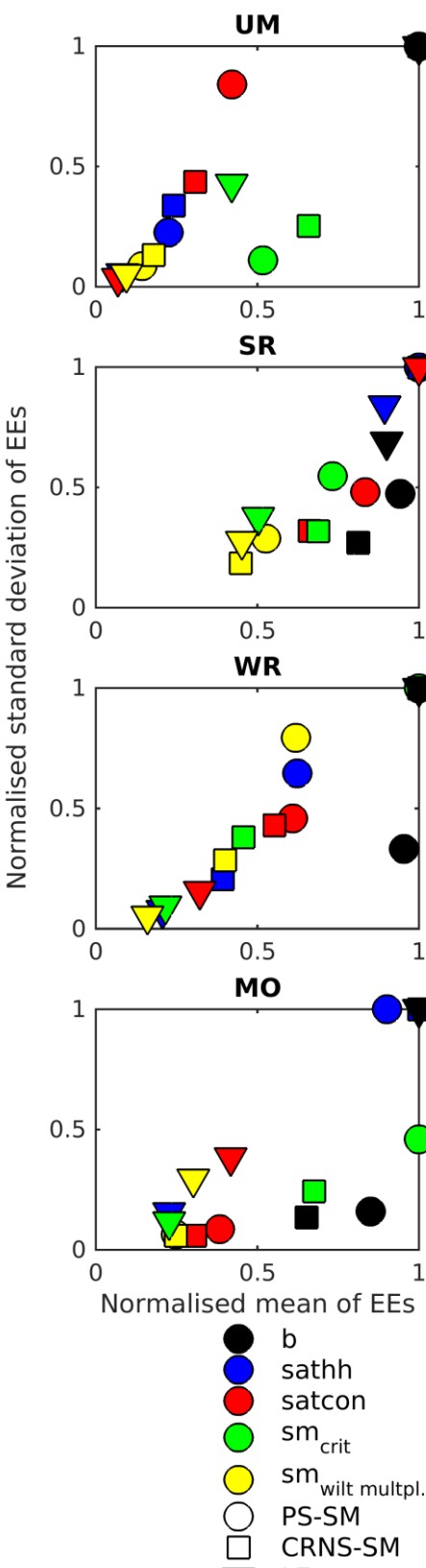

**Figure A2.1: Morris sensitivity indices for the five calibrated parameters at four sites: UM, SR, WR, and MO. The three Objective Functions are the RMSEs between simulated and observed PS soil moisture (PS-SM), CRNS neutron counts (CRNS-N), and latent heat flux (LE). The mean elementary effects (EEs), representing the main effects, are displayed along the horizontal axis. The standard deviations of the elementary effects, representing parameter interactions, are shown along the vertical axis. Values were normalised to the most influential parameter for each Objective Function.**