# Peer review of "Land surface model performance using cosmic-ray and point scale soil moisture measurements for calibration"

_Hydrology and Earth System Sciences, 2016_

## Referee Comment (RC1) · Anonymous Referee #1 · 6 Dec 2016

OVERVIEW:

The presented manuscript investigates the potential impact of the measurement scale for calibration of a land surface model. For this purpose, observed and simulated land surface data at 12 sites on the continental US from several sources including Ameriflux, COSMOS and NLDAS was used. Point scale soil moisture data was compared to cosmic ray soil moisture retrievals. Furthermore, land surface simulations at the nine sites were done on an individual basis using JULES. At each sites, JULES was calibrated with cosmic ray data, point scale soil moisture data and eddy flux measurements. Model results were evaluated with eddy flux and soil moisture measurements. The case study demonstrates the added value of cosmic ray measurements at the model scale compared to local scale soil moisture measurements.

[Figure]

However, the study needs a major revision that addresses readability in the following: Reduce/clarify abbreviations, restructure part of the manuscript, improve English / sentence structure, remove speculations, be more specific /quantitative at a number of instances. There seems to be an issue with the data presented in Figure 7 concerning site MO.

The figures require further formatting. I suggest reducing the number of Figures. This allows the reader to focus on the essential messages of the study. I disagree with the outcome that coupling of soil moisture and latent heat flux is weak in JULES (e.g. see comment to Figure 9). Further suggestions in the Specific comments.

GENERAL COMMENTS:

The paper exhibits a clear novelty by quantifying the impact of using cosmic ray soil moisture data for calibration as compared to local point soil moisture measurements. The study fits the scope of the HESS journal and deserves to be published in HESS after major revision.

The conclusions reached in the manuscript are not clear enough. I also found different conclusions from the data and results presented. For details: See Specific comments to Chapter 4. The scientific methods and assumptions were well chosen and represent state of the art.

Description of experiments and calculations need to be revised. I suggest following new structure: Chapter 2.1 can remain there. Then, Chapter 2.3 should be changed to Chapter 2.2 as soil moisture data should be compared before calibration or modeling. Then explain JULES, then JULES forcing and initial conditions, the following Chapters can remain in place.

The results chapter needs a new structure. The results are presented in the right order but intermittent by discussions that are out of place because there IS a Chapter "3.8 Discussion". A more clear structure would be either consistently "3. Results and

Discussion" or "3. Results; 4. Discussion; 5. Conclusion". Please, stick to either one but do not mix.

The topic is complex and in general well addressed, but a new structure will increase readability and will make writing the paper more easy.

The title reflects the content. However, I would suggest a modification of the title to e.g. "Improved land surface processes by calibration with cosmic ray soil moisture measurements at the model scale".

The abstract is concise and summarizes the paper well. It may be modified if conclusions are changed.

In general, there is a large number of abbreviations (e.g. PS), symbols (in formulas), short names (e.g. smcrit). This makes the paper very difficult to follow. It is necessary to use either abbreviations, or symbols also in the text, ommit short names and in general write the names out more often. This paper almost needs a List of Abbreviations. Please, reduce them.

SPECIFIC COMMENTS:

Page 1 line...: 9: "sometimes" is too unprecise. Rephrase e.g. "can be calibrated" 10: EC, PS, LSM, CRNS, JULES - How can the abbreviations in the abstract be reduced? 12: Is there a term "soil-evapotranspiration"? I suggest "soil and evapotranspiration" or "soil-evaporation". 17: "CRNS calibrations" – What is actually meant is "LSM calibration" or something alike. 30: "atmospheric circulation" - In the UK? Be more specific here or do not mention it. 30: "Because", is "because" naturally a start of the sentence?

Page 2 line...: 1: "believed" - It is for THESE authors important or it is not. Maybe, why is it important. There is no "believe" in science. 4: "processes." Reference missing. 4: Remove "however". no added value. 14-15. Unclear, specify or rephrase. 22: "in-situ soil moisture" – I suggest to be consistent throughout the paper. in-situ or point scale is the same. So I suggest using one phrasing only. 24: soil moisture "IS" spatially...

and add reference e.g. Qu et al. 2015 in GRL. 26-28: "EC footprint average soil moisture" or "LSM grid cell" average soil moisture depends on the objective. As is, it is confusing. Rephrase to be clear. 31: Is this the research question? There is another research question on the next page. Either both should be allocated together or only one objective / research question is needed. As is, it is confusing.

Page 3 line...: 5 "usually assumed" - this is rather a fact due to soil heterogeneity. 10: One sentence paragraphs should be avoided. Also the use of "which" as often as it is used in the present manuscript, should be avoided. In English, short sentences are much better understood and much clearer. One sentence is preferably 1-2 lines only. 12: add the German CRNS network (Baatz et al 2015) 13: Repetition. 16: "similar" - be more specific. Similar is least informative and makes room for interpretation. 17: which – split the here sentence. 21: which - rephrase to "These sites" also look up the difference between which and that. Is the information you give with "which" really necessary? Then put it in one single sentence as it is worth a single sentence. 27: that – could that be removed here? 30: "Before our modelling exercise" ... This sentence should be moved up within the paragraph. First things first. 30-33: A shorter sentences are desirable.. It can be two or more sentences.

Page 4 line...: 10: "used data" - Specify "data". 10: remove brackets and specify e.g. "the upper first 30 cm". 15: Remove "Similar". 18: "CRNS integrated soil moisture" ... rephrase for better understanding e.g. soil moisture integrated over depth from CRNS soil moisture, hereafter referred to as CRNS soil moisture retrieval. 23: Split sentence, remove while. 31: remove which, split sentence 32: "More than 31 days were gap filled" using average diurnal pattern. This sounds like a really high uncertainty. Is this the case for precipitation, too? Is the high uncertainty reflected in the results? If so, where? Is it feasible to mark this in the Figures? Is it feasible to remove these periods from the calibration period? How much of modeled periods was filled with diurnal patterns?

Page 7 line...: 8: Rather "Calibration Approaches". Where is the "two-objective calibration" in the methods section? 10-12: Sentence too long. Rephrase into at least two sentences. 12: Add reference (Shuttleworth et al . 2013) 14-15: This reads difficult. Either write saturated hydraulic conductivity instead of sathh or use the symbol introduced in Eq. 5. Same for all other symbols throughout the manuscript. 22-25: Shorter sentences. 27: This this.... rephrase

Page 12 line ...: 32: Remove "It should be noted however that".

Page 13 line...: 6: I suggest to put "and also containing stones" to the end of sentence or another place and remove brackets. 7: Here you write sathh out. Much better. However, the sentence may be moved to the discussion section. 13: I suggest to discuss these reasons. Actually, it is worth investigating each of the points to either accept them or rule them out. Mentioning all of these points / reasons is not getting the manuscript closer to the objective. 35: The "two-objective calibration" was not mentioned before. However, it is good to have it.

Page 14: line...: 5: comma before respectively. 7: was instead of "could have been" 7: "similar" - quantify or remove. 11: Suggestion: Move "to have obtained such automatic improvement" to the end of sentence 16: "improved soil moisture and latent heat flux". If calibration is done for both, both should be improved, no? 18: What does EF stand for? Too many abbreviations. Also WO...etc. There should be a way to distinguish places/sites from variables from acronyms. 19: "coincidence" – Not really, is it? 20: "Similar" - be precise. 21: "this" means what? context not totally clear. Rephrase. 27: "Generally quite weak" - Reference in literature? I cannot quite follow here. 28: What is the reasons that at these sites strong coupling is expected? At what time? Is the calibration done during the time of strong coupling? Very unclear how this conclusion is reached. I suggest an individual point of discussion. This is also the point where I cannot follow the conclusion drawn. It would be a major setback of JULES which needs to be justified much stronger. Above all, the results in the Figures show a meaningful difference in ET due to calibration / calibration method.

[Figure]

Page 15 line...: 4: "day time only" Why suddenly "day time only"? Is this throughout the manuscript the case or just here? LE measurements are difficult in the night. Diurnal cycles are difficult to obtain. Only modeled LE at times of observed ET should be compared, because there is no observations at other times. I do not see this here. 15: "Weak coupling..." Can the forcing data be the reason here? I disagree, see comments to Figure. 16: "CRNS-N/LE" - rephrase and make clear. 20-21: Suggestions: Split sentence into two. 28: "root zone soil moisture" is where exactly and calculated how? 32: split sentence at "while" and remove "while". Also at other instances in the manuscript, this makes sence.

Page 16 line ...: 4: Again the relatively weak coupling is discussed. Avoid repetition. Restructure. 5: "this suggests that how" - rephrase 9: Parameters should not move up or down after calibration. Variables and model states may be variable. "move closer" to what? 10: Which implications? Please name them and argue why. 14: "did not differ substantially". I disagree. They did differ. 17: "could be used"... you can put "should" instead. "we might have found worse fits". This is speculations. Please give reason or remove speculations. One could state: "You might have found better fits." 21: "Our findings support this." Your findings were that the JULES model is not an "improved land surface model", so the manuscript as is cannot support this. However, I see an impact of soil moisture states on latent heat flux in your model runs. Just have a look on Figure 11, how the RMSE in LE is reduced by calibration at sites DC,SO,KE,SR,WR. Only at few sites, RMSE in LE became higher. Soil moisture seemed to impact latent heat flux in JULES. The authors can be and should be more positive in the results, discussion and conclusion.

Table 1: -

Table 2: Add symbols from the equations and use symbols within the text.

Figure 1: I suggest removing the figure. The scale mismatch can be pointed out in a single sentence to save space for result figures e.g. The model grid cell size is 1km,

the EC footprint is between 100m2 to 1km2 (put here diameter instead), the CRNS footprint is 300m in diameter, the PS soil moisture sensor measures few dm3.

Figure 2: This Figure is very informative and necessary. Put Figure 2a and 2b instead of "upper".

Figure 3: Informative, you may want to keep it, however consider removing.

Figure 4: Here, MO looks like it has a big discrepancy / bias in PS to CRNS soil moisture. Why is this the case / is this in the manuscript? Put the legend (PS, CRNS)beside or below the figure, not in the first subplot. Minimum of Y axis is missing.

Figure 5: Informative. Keep it. CRNS observations are neutron flux. Soil moisture is retrieved from CRNS neutron flux observations. It should be rather something like CRNS soil moisture retrieval than CRNS soil moisture observation. Again, SM is abbreviated in your figure as SM, but not within the paper or at few instances. In general, abbreviations should be reduced.

Figure 6: What is the added value to Figure 5, because JULES is calibrated based on the data used in Figure 5. I suggest to merge Figures into one with e.g. Figure 5a and 5b.

Figure 7:Add grid lines. How can CRNS and PS soil moisture at MO be so different as in Figure 5, but Model results after calibration be so similar as in Figure 7. It seems very strange. Also you plot a 3 year time series with hourly values. It will be much easier to read, with more information and will have even more meaning if you average over days or months. So far, it is a lot of variability, which is clear beforehand if hourly values are plotted over the course of 3 years. You may consider merging Figure 7 and Figure 4.

Figure 8: The indices a) and b) are missing in the Figure. Axis labels at wrong place. Consider merging with Figure 10,11,12. The data may also fit into one table.

Figure 9: Interestingly WR and MO show a really strong change from Default to Calibrated. How do you get to the conclusion that there is no coupling of LE to soil moisture in JULES?

Figure 13: Labels and legend are overlapping.

Baatz, R., H. Bogena, H. J. Hendricks-Franssen, J. A. Huisman, W. Qu, C. Montzka, and H. Vereecken (2014), Calibration of a catchment scale cosmic-ray probe network: A comparison of three parameterization methods, J Hydrol.

Shuttleworth, J., R. Rosolem, M. Zreda, and T. Franz, 2013: The COsmic-ray Soil Moisture Interaction Code (COSMIC) for use in data assimilation. Hydrol Earth Syst Sc, 17, 3205-3217.

W. Qu, H. R. Bogena, J. A. Huisman, J. Vanderborght, M. Schuh, E. Priesack, H. Vereecken, Predicting subgrid variability of soil water content from basic soil information, Geophysical Research Letters, 2015, 42, 1-8.

---

## Referee Comment (RC2) · Dr. Teuling (Referee) · 2 Jan 2017

**1    General comments**

The manuscript by Iwema et al. addresses the use of new Cosmic-Ray Neutron Sensor (CRNS) data in land surface model calibration.  By using CRNS and in situ sensor data from 12 sites across the U.S., the authors systematically investigate to which degree calibration against this novel data leads to an improvement in the simulation of observed surface fluxes.  Unfortunately the results do not show a clear improvement of using CRNS data over in situ data, but this does not in any way affect the quality of the research.  Overall, I have a positive impression of the manuscript which I believe would make a good contribution to HESS, but it will need improvements on several

aspects. In particular, some of the main conclusions do not seem to follow from the results (a problem also identified by the other referee), the authors should make clear that the in situ data is not used to its full potential so the study is not a clean comparison between datasets but rather a comparison of scale, and the authors seem to have been somewhat selective in the selection of references to identify the knowledge gap. These issues are discussed in more detail below. However I believe these comments can be addressed by minor (mostly textual) revisions.

A first remark concerns the title, which does not seem to reflect the contents of the manuscript. The study really looks at calibration using soil moisture observations at different scales, and it does not look at measurement scales as such. I suggest a title along the lines of "Land surface model performance using cosmic-ray and in situ soil moisture data for calibration". Also, the use of "reducing" is seemingly at odds with the results, which does not show an improvement when using observations at the "model" scale.

As was also pointed out by the other referee, the conclusion that JULES has a weak coupling between soil moisture and ET does not follow from the results. In fact, JULES has a strong coupling by definition, and the coupling in reality can only be less strong. The fact that ET estimates do not improve with improvements in soil moisture is likely because ET is not so sensitive to changes in soil parameters, even for different climate conditions. This is for instance shown in a paper I wrote in 2009 (Teuling et al., 2009), in which I investigated effects of soil parameters on soil moisture and ET. In short, effects of soil parameters on soil moisture are generally large, but effects on ET are small. This is primarily caused by the main effect of soil parameters which is a shift in the mean, rather than the dynamics (this is also consistent with the strong contribution of bias to MSD as reported by the authors). This should be discussed better.

Related to this is the question why rooting depth was not optimised along with the soil parameters. If the model rooting depth does not reflect the actual root profile, optimization of soil parameters along will not lead to a better estimation of available

soil water. These choices should be better motivated.

In making a case for the validity of their research question, the authors miss out on an important body of literature on spatio-temporal characteristics of soil moisture fields. It has been shown by numerous studies that while soil moisture generally shows a large spatial variability, individual points maintain their rank while the mean changes. This insight started with the "classical" Vachaud et al. study, but numerous other studies (for instance Teuling et al., 2006, Mittelbach and Seneviratne, 2012, among others) have reported similar behaviour. This behaviour implies a relatively small spatial variability of fluxes, which was explained from a theoretical perspective by Albertson and Montaldo (2003) and explored Teuling and Troch (2005), among others. In effect, based on these studies, the hypothesis could also be formulated more neutral in the form of a null-hypothesis: "A reduced scale mismatch does not lead to LSM flux estimates closer to eddy covariance observations..". The results could subsequently be interpreted as insufficient evidence to reject this hypothesis. In any case the introduction should be changed to include a discussion on the relation between point-scale and large-scale soil moisture. This can then also be used to explain the reference to Franz et al. (2012), which now conflicts with the hypothesis (if CRNS and in-situ soil moisture have already been reported to be similar, how can the authors still expect to find a difference in fluxes?). These comments do not make the research any less relevant, but the phrasing of the hypothesis should be in line with the "state-of-the-art", and not a convenient selection thereof.

In applying CRNS and PS soil moisture products, the authors only consider the shallower PS soil moisture observation in order to comply with the CRNS observation depth. While this makes sense given the goal of the study (scale mismatch and not a comparison of observation techniques), it is not sufficiently recognized that this introduces an unfair disadvantage to the PS data. There might be important information in deeper PS observations, in particular during soil moisture-limiting conditions, that is not being considered in this study. It is thus crucial to make a clear distinction between the

scale aspect and the observation technique, and acknowledge in the discussion in the discussion that PS soil moisture might give better results when all available observations are used. A better alternative would be to redo the analysis using all observation depths in addition to using only the shallow observations, but this would likely require a substantial amount of work.

**2  Detailed comments**

Page 1, Line 27: it is not the coupling between soil moisture and flux partitioning that is strong (this is similar across climates), but the soil moisture control on temperature at seasonal timescales.

Page 3, Line 15: If the CRNS data is known to give similar values as the in situ data, how can this be consistent with the hypothesis posed later on? Please change the line of reasoning, but also take into account my previous comments on missing references on soil moisture temporal stability.

Page 5, Line 28: Richards equation -> Richards' equation

Page 13, Line 13: "We could think of..." ->All arguments are backed up by references, so it might be better to say "Several reasons have been proposed..."

Figures: Generally ok, but I agree with the comments of the other referee that the number can be significantly reduced without sacrificing the main message.

**3  References**

Albertson and Montaldo, 2003. Temporal dynamics of soil moisture variability: 1. Theoretical basis, Water Resour. Res., 39, 1274.

Mittelbach and Seneviratne, 2012. A new perspective on the spatio-temporal variability of soil moisture: temporal dynamics versus time-invariant contributions. Hydrol. Earth Syst. Sci. 16(7), 2169-2179.

Teuling et al., 2006. Estimating spatial mean root-zone soil moisture from point-scale observations. Hydrol. Earth Syst. Sci. 10, 755–767.

Teuling et al, 2009. Parameter sensitivity in LSMs: An analysis using stochastic soil moisture models and ELDAS soil parameters. J. Hydrometeorol., 10(3), 751-765.

Vachaud et al., 1985. Temporal stability of spatially measured soil water probability density function, Soil Sci. Soc. Am. J., 49, 822–828.

---

## Referee Comment (RC3) · Anonymous Referee #3 · 3 Jan 2017

**OVERVIEW**

The manuscript investigates the use of different in situ soil moisture datasets for improving surface energy flux estimation from land surface modelling. Specifically, the JULES land surface model is calibrated against point scale (PS) and cosmic-ray neutron sensor (CRNS) soil moisture data. The rationale is that CRNS provide measurements at larger scale than PS observations and, hence, they are more appropriate for surface energy fluxes estimation.

**GENERAL COMMENTS**

The manuscript is quite well written and clear, even though some parts should be re-
duced and summarized. The topic is surely of interest for the HESS readership as cosmic-ray probes represent a new technology for ground measuring soil moisture over large areas. Therefore, we need to assess the impact of this new technology for improving land surface modelling. The paper describes the calibration of JULES land surface model with PS and CRNS at different sites in US. The manuscript is well conceived and applied over a large number of sites thus obtaining reliable and robust results. However, I mostly agree with the comments of previous reviewers, and particularly I believe that several aspects should be improved/changed before the publication. I reported below a list of the general comments to be addressed with also the specification of their relevance.

1) **MAJOR**: I found some of the explanations/justification of the results given in the paper quite weak. They appear to me as speculations, not supported by the performed analyses and results. For instance, I refer to:

(A) The comparison between PS and CRNS soil moisture data (section 3.1.3) shows that soil moisture timeseries are quite similar. The authors expected better performances at homogeneous sites but it was not the case. As shown in "temporal stability" papers (see also Teuling report), PS measurements are usually very well correlated with large scale measurements. Therefore, I expect good correlations. In my opinion, the good (bad) performances are mostly related to the good (bad) quality of soil moisture observations that may be affected by a number of factors (e.g., soil texture, sensor malfunctioning, ...). Therefore, theoretically I could expect that CRNS are better than PS measurements, but due to measurement uncertainties and errors, the larger support scale of CRNS is masked out by the (likely) lower quality of their measurements. This important aspect, i.e., the quality of soil moisture observations, should be carefully addressed in the paper.

(B) The authors attributed the low differences in estimating surface energy fluxes when PS and CRNS measurements are considered to the weak coupling in JULES between soil moisture and evapotranspiration. Actually, I do not believe it is the case, but it

should not be a speculation. It should be tested. If the authors want to give this message, they should demonstrate that with a different model or land surface scheme the differences are higher, and likely CRNS is better that PS data (as expected at the beginning). Therefore, I suggest changing the conclusions or, better, implementing an additional LSM and demonstrate the results through a scientifically sound approach.

(C) The range of reasons reported at page 13, lines 13-34 are only speculations. I suggest removing.

2) **MAJOR**: I found quite strange that by using the default parameter values performs the same than using the parameter values calibrated on soil moisture data in terms of evaporation fraction (EF) estimation. Even though soil moisture data were of low quality, or the coupling between soil moisture and EF is weak, soil moisture observations represent local data that should give some information to the model. Therefore, I expected better results with respect to the default parameterization. What happens if JULES is calibrated on EF data? How the corresponding modelled soil moisture data compare with PS and CRNS observations? By looking at the results reported in the paper, it seems that using soil moisture observations is needless if we have the purpose of improving land surface modelling. I suggest the authors to improve the discussion and the analysis of the results.

3) **MODERATE**: I found the description of the results with too many details in several parts of the text(e.g., section 3.1.1, page 9 lines 14-28, page 12, lines 3-18). I suggest not discussing the results for each site, but trying to summarize the most important findings and to focus the discussion on these results.

In the specific comments, I added some corrections and suggestions that should be implemented.

On this basis, I believe the paper deserves to be published only after a major revision.

**SPECIFIC COMMENTS (P: page, L: line or lines)**

P2, L26: The sentence "past research indicates ...wetting and drying periods" is too vague. At least, references should be included. However, I note that it is still an open issue to fully understand in which conditions soil moisture variability is higher. For instance, it is not the same if absolute or anomaly soil moisture values are analysed (see e.g., Mittelbach and Seneviratne, 2012; doi:10.5194/hess-16-2169-2012).

P3, L8-10: The sentence is incomplete (only a single sensor is needed for?), please check.

P4, L32: The gap-filling of 30 days seems to me a very large window. Does it affect the results? Some tests should be made.

P5, L6-10: It should be better to insert an equation here.

P9, L20: The larger differences between PS and CRNS at wetter sites are expected. Higher is soil moisture, higher will be the differences.

P10, L5: Figure A1 should be Table A1?

P10, L18-22: The difference in sensing depth might be the cause of some of the differences between PS and CRNS. However, it could be checked with specific analysis. Otherwise, I suggest removing.

P10, L29: It is obvious that after the calibration on soil moisture data the RMSE values will reduce. The model is tested with the data used for calibration.

P11, L1-2: The use of RMSE for calibration reduces the mean error between modelled and observed data. Therefore, the effect of having less extreme peaks and valleys is due to the selected objective function.

[Figure]

P11, L24-28: From here it is not clear the number of sites over which an improvement in EF estimation is obtained. 11 sites (P11, 22) or 12 sites (P11, L28). Check also later in the text (e.g., P13, L36).

P12, L3-6: It is not clear to me what the authors want to demonstrate with this analysis. Please clarify.

P12, L32: Change "on the edges" with "within the edges".

P15, L9-12: Not clear to me how the "multiplier" values are used. Please clarify.

P16, L4-11: As mentioned above, I found not scientifically sound to attribute the low performances in term of EF improving to the weak coupling of JULES. Moreover, it's not clear to me the discussion of the value of absolute soil moisture with respect to anomalies.

Figure 8: Labels a) and b) are missing.

―――――――――――――――――――

---

## Author Comment (AC1) · 10 Jan 2017

Answer to interactive comment by Anonymous Referee #1

We would like to thank the Referee for carefully reviewing the manuscript. An initial reply to the reviewer's comments is provided below following the original review comments shown in Italic font.

*OVERVIEW*

*The presented manuscript investigates the potential impact of the measurement scale for calibration of a land surface model. For this purpose, observed and simulated land surface data at 12 sites on the continental US from several sources including Ameriflux, COSMOS and NLDAS was used. Point scale soil moisture data was compared to cosmic ray soil moisture retrievals. Furthermore, land surface simulations at the nine sites were done on an individual basis using JULES. At each sites, JULES was calibrated with cosmic ray data, point scale soil moisture data and eddy flux measurements. Model results were evaluated with eddy flux and soil moisture measurements. The case study demonstrates the added value of cosmic ray measurements at the model scale compared to local scale soil moisture measurements.*

*However, the study needs a major revision that addresses readability in the following: Reduce/clarify abbreviations, restructure part of the manuscript, improve English / sentence structure, remove speculations, be more specific /quantitative at a number of instances. There seems to be an issue with the data presented in Figure 7 concerning site MO.*

*The figures require further formatting. I suggest reducing the number of Figures. This allows the reader to focus on the essential messages of the study. I disagree with the outcome that coupling of soil moisture and latent heat flux is weak in JULES (e.g. see comment to Figure 9). Further suggestions in the Specific comments.*

ANSWER:
We thank the referee for this evaluation of the manuscript. As also pointed out by other reviewers, we will address the issues mentioned above by the referee to increase the manuscript readability. The issue with Figure 7 is explained below in the 'specific comments'. We agree that our use of the word coupling was not appropriate as also pointed out by the other reviewers. Our interpretation is, in fact, in line with Reviewer #2 comments which should emphasize that the results obtained in our study are a consequence of calibrating soil parameters, rather than the soil moisture – evapotranspiration coupling in JULES.

*GENERAL COMMENTS:*
*COMMENT: The paper exhibits a clear novelty by quantifying the impact of using cosmic ray soil moisture data for calibration as compared to local point soil moisture measurements. The study fits the scope of the HESS journal and deserves to be published in HESS after major revision.*

*The conclusions reached in the manuscript are not clear enough. I also found different conclusions from the data and results presented. For details: See Specific comments to Chapter 4. The scientific methods and assumptions were well chosen and represent state of the art.*

ANSWER: We thank the reviewer for his/her positive comments regarding our manuscript and its relevance to the HESS community. We will address the issues regarding the clarity and inconsistency of our conclusions in the revised version as pointed out in detail by the reviewer's comments.

*COMMENTS:*

*Description of experiments and calculations need to be revised. I suggest following new structure: Chapter 2.1 can remain there. Then, Chapter 2.3 should be changed to Chapter 2.2 as soil moisture data should be compared before calibration or modeling. Then explain JULES, then JULES forcing and initial conditions, the following Chapters can remain in place.*

*The results chapter needs a new structure. The results are presented in the right order but intermittent by discussions that are out of place because there IS a Chapter "3.8 Discussion". A more clear structure would be either consistently "3. Results and Discussion" or "3. Results; 4. Discussion; 5. Conclusion". Please, stick to either one but do not mix.*
*The topic is complex and in general well addressed, but a new structure will increase readability and will make writing the paper more easy.*

ANSWERS: We will revise the structure of our manuscript appropriately to improve its readability and clarity. We thank the referee for his/her suggestions.

*COMMENT: The title reflects the content. However, I would suggest a modification of the title to e.g. "Improved land surface processes by calibration with cosmic ray soil moisture measurements at the model scale".*

ANSWER: We thank the reviewer for making this point which was also raised by Referee #2. We are considering the following new title: "Land surface model performance using cosmic-ray and point scale soil moisture data for calibration".

*COMMENT: The abstract is concise and summarizes the paper well. It may be modified if conclusions are changed.*

ANSWER: We appreciate that the Referee found our abstract to be concise and a good summary of our manuscript

*COMMENT: In general, there is a large number of abbreviations (e.g. PS), symbols (in formulas), short names (e.g. smcrit). This makes the paper very difficult to follow. It is necessary to use either abbreviations, or symbols also in the text, ommit short names and in general write the names out more often. This paper almost needs a List of Abbreviations. Please, reduce them.*

ANSWER: This is a valid point raised by the referee. Whenever possible, we will reduce the number of abbreviations and/or include a List of Abbreviations for clarity. Notice that some abbreviations are already widely accepted by the scientific community (e.g., LSM, CRNS) and we believe in those cases, the use of them are justified.

*SPECIFIC COMMENTS*
The referee has made a number of suggestions to significantly improve the quality of the manuscript, in particular regarding to its structure, clarity, and use of language. We thank the Referee for his comments and will make appropriate changes in the revised version. Below, we focus on the main comments that concern with the methodology and conclusions drawn.

*COMMENT:*
*Page 4, line 32: "More than 31 days were gap filled" using average diurnal pattern. This sounds like a really high uncertainty. Is this the case for precipitation, too? Is the high uncertainty reflected in the results? If so, where? Is it feasible to mark this in the Figures? Is it feasible to remove these periods from the calibration period? How much of modeled periods was filled with diurnal patterns?*

ANSWER:
We thank the reviewer for raising this point. This was also mentioned by reviewer #3. We provide a table below summarising the average (average of the seven forcing variables) percent gap for each site. Precipitation was not gap-filled; missing points were set to zero instead. Overall, average gaps vary among sites between near zero to 15%. As pointed out by the reviewer, this can introduce some uncertainty in the analysis and we will highlight data when describing the dataset used in this study. We will also mentioned in the text the original gap period in the data.

| Site | Percentage missing hours filled: mean (range) | Size of time series in years |
|------|-----------------------------------------------|------------------------------|
| UM | 7 (2-12) | 1.5 |
| DC | 1 (0-5) | 3.7 |
| SO | 7 (0-15) | 3.8 |
| KE | 1 (0-3) | 4.6 |
| ME | 0 (0-0.1) | 1.6 |
| SR | 2 (0-14) | 3.6 |
| CS | 2 (0-5) | 3.7 |
| MM | 1 (0-2) | 2.7 |
| TR | 0.1 (0-0.2) | 3.6 |
| AR | 10 (8-14) | 2.5 |
| WR | 2 (0-4) | 2.6 |
| MO | 3 (1-7) | 2.7 |

*COMMENT:*
*Page 13, line 13: I suggest to discuss these reasons. Actually, it is worth investigating each of the points to either accept them or rule them out. Mentioning all of these points / reasons is not getting the manuscript closer to the objective. 35: The "two-objective calibration" was not mentioned before. However, it is good to have it.*

ANSWER:
Page 14, line 13: We thank the referee for his/her comments. Note, this issue was also raised by Referee #3 which suggested for removal. We re-assess this section of the manuscript for the revised version of the manuscript.

*COMMENT:*
*Page 14, line 19: "coincidence" – Not really, is it?*

ANSWER:
Page 14, line 19: We understand that the word "coincidence" may not have been chosen appropriately in this context. We explain our observation better in the revised version of the manuscript. What we meant to say is that there is: A small deterioration in PS soil moisture or CRNS neutron simulation could yield a much worse latent heat flux simulation, even of the same order as the default run. This happened for example for the CRNS calibration at SR (Figure 11), where a deterioration of only about 0.05 in the normalised RMSE-neutron count could have yielded the same normalised RMSE-LE as the default run (1). This implies a large (e.g. 0.8) improvement in soil moisture (in terms of in normalised error) would not necessarily mean an improvement in latent heat flux.

*COMMENT:*
*Page 14, line 28: What is the reasons that at these sites strong coupling is expected? At what time? Is the calibration done during the time of strong coupling? Very unclear how this conclusion is reached. I suggest an individual point of discussion. This is also the point where I cannot follow the conclusion drawn. It would be a major setback of JULES which needs to be justified much stronger. Above all, the results in the Figures show a meaningful difference in ET due to calibration / calibration method.*

ANSWER:
Page 14, line 28: As mentioned above, we will improve the discussion to appropriately address the issue of consequence of soil parameters calibration rather than strong/weak coupling as originally stated in the manuscript.

*COMMENT:*
*Page 15 line...: 4: "day time only" Why suddenly "day time only"? Is this throughout the manuscript the case or just here? LE measurements are difficult in the night. Diurnal cycles are difficult to obtain. Only modeled LE at times of observed ET should be compared, because there is no observations at other times. I do not see this here.*

ANSWER:
Page 15 line 4: This is an important point raised by the referee. Latent heat flux and evaporative fraction performance were computed over day time values only (Page 8 line 16). Notice that night time values were used to plot the monthly mean diurnal cycles shown in Figure 9, but did not make any contribution to the computation of the validation metrics (RMSEs). We will make this clearer where we present the results and in the figure captions.

*COMMENT:*
*Page 15, line 28: "root zone soil moisture" is where exactly and calculated how?*

ANSWER:
Page 15, line 28: JULES does not provide a root zone soil moisture output weighted by the presence of roots that can be directly compared with the values of the wilting point and critical point soil moisture. Therefore we computed an estimation of root zone soil moisture by computing the soil moisture stress weighting factor for each layer at each time step (Equation 3). We then computed the relative contribution from each layer to the total root zone soil moisture stress factor. This relative contribution is a function of the root density and soil thickness. To finally obtain the weighted root zone soil moisture we multiplied the JULES soil moisture of each layer with the weighting factor.

*COMMENT:*
*Page 16, line 14: "did not differ substantially". I disagree. They did differ.*

ANSWER:
Page 16, line 14: We argue there was no substantial difference between calibrating against PS and against CRNS data with respect to latent heat flux across sites. We base this on the results from the two-objective calibrations against soil moisture and latent heat and against neutron counts and latent heat (Figure 12). Only at SR (CRNS better), DC (CRNS better), TR (PS better), and UM (PS better) did we observe a substantial (larger than 10%) difference between PS and CRNS calibration, based on the compromise solutions from the two-objective calibrations. To increase readability, we consider using latent heat as principle validation metric instead of evaporative fraction. The results for evaporative fraction (figure 8) would then be moved to the appendix, while Figure A2.1 would become the new Figure 8 (and a scatter plot like current Figure 8b will be added.

*COMMENT:*

*Page 16, line 21: "Our findings support this." Your findings were that the JULES model is not an "improved land surface model", so the manuscript as is cannot support this. However, I see an impact of soil moisture states on latent heat flux in your model runs. Just have a look on Figure 11, how the RMSE in LE is reduced by calibration at sites DC,SO,KE,SR,WR. Only at few sites, RMSE in LE became higher. Soil moisture seemed to impact latent heat flux in JULES. The authors can be and should be more positive in the results, discussion and conclusion.*

ANSWER:

Page 16, line 21: This is a very important point mentioned by the reviewer. We will re-phrase our conclusions on JULES because, as the reviewer mentions, JULES soil moisture does affect simulated surface energy partitioning and latent heat flux. The authors do still not see a substantial difference in surface energy partitioning and latent heat flux simulation when CRNS neutron counts are used instead of PS soil moisture. Our revision will focus on the limited effects of calibrating soil parameters on evapotranspiration as pointed out by Referee #2 based on his previous work (e.g., Teuling et al., 2009). We will also expand our analysis to incorporate discussions on soil moisture data quality from both measurement approaches (as pointed out by Referee #3) and expected behaviour from spatio-temporal stability (as pointed out by Referee #2).

*COMMENT:*

*Figure 7: How can CRNS and PS soil moisture at MO be so different as in Figure 5, but Model results after calibration be so similar as in Figure 7. It seems very strange.*

ANSWER:

Figure 7: Please notice that the overall calibration based on PS and CRNS (i.e., top and bottom MO panels in Figure 7) actually yielded different results. The figure on page 6 of this answer combines both plots to highlight this fact (notice default run line is omitted as it is irrelevant to this comparison).

[Figure]

**Hourly observed (PS and CRNS) soil moisture time series and simulated soil moisture time series after calibration against PS and CRNS soil moisture for site MO.**

*COMMENT:*
*Figure 9: Interestingly WR and MO show a really strong change from Default to Calibrated. How do you get to the conclusion that there is no coupling of LE to soil moisture in JULES?*

ANSWER:
Figure 9: The Referee is right that WR and MO show a strong change from default to single-objective calibrated. As mentioned above, we will improve the discussion to appropriately address the issue of consequence of soil parameters calibration rather than strong/weak coupling as originally stated in the manuscript.

REFERENCES
Teuling et al, 2009. Parameter sensitivity in LSMs: An analysis using stochastic soil moisture models and ELDAS soil parameters. J. Hydrometeorol., 10(3), 751-765.

---

## Author Comment (AC2) · 10 Jan 2017

Answer to interactive comment by Referee #2 Dr. Teuling

The authors thank Dr. Teuling for providing comments and suggestions, which will help us to improve our manuscript. An initial reply to the reviewer's comments is provided below following the original review comments shown in Italic font.

*1. General comments*

*COMMENT:*
*The manuscript by Iwema et al. addresses the use of new Cosmic-Ray Neutron Sensor (CRNS) data in land surface model calibration. By using CRNS and in situ sensor data from 12 sites across the U.S., the authors systematically investigate to which degree calibration against this novel data leads to an improvement in the simulation of observed surface fluxes. Unfortunately the results do not show a clear improvement of using CRNS data over in situ data, but this does not in any way affect the quality of the research. Overall, I have a positive impression of the manuscript which I believe would make a good contribution to HESS, but it will need improvements on several aspects. In particular, some of the main conclusions do not seem to follow from the results (a problem also identified by the other referee), the authors should make clear that the in situ data is not used to its full potential so the study is not a clean comparison between datasets but rather a comparison of scale, and the authors seem to have been somewhat selective in the selection of references to identify the knowledge gap. These issues are discussed in more detail below. However I believe these comments can be addressed by minor (mostly textual) revisions.*

ANSWER:
We thank the Referee for his effort and for his positive comments regarding the quality and relevance of our study to the HESS community. We will revise the conclusions, so they do follow from the results, as also provided in the answer to Referee #1 and Referee #3. We will also appropriately discuss the relevant points based by the referee in the revised manuscript and include additional supporting literature as mentioned.

*COMMENT:*
*A first remark concerns the title, which does not seem to reflect the contents of the manuscript. The study really looks at calibration using soil moisture observations at different scales, and it does not look at measurement scales as such. I suggest a title along the lines of "Land surface model performance using cosmic-ray and in situ soil moisture data for calibration". Also, the use of "reducing" is seemingly at odds with the results, which does not show an improvement when using observations at the "model" scale.*

ANSWER:
We thank the Referee for making this valuable point. We will change the title appropriately to reflect the reviewer's suggestion.

*COMMENT:*
*As was also pointed out by the other referee, the conclusion that JULES has a weak coupling between soil moisture and ET does not follow from the results. In fact, JULES has a strong coupling by definition, and the coupling in reality can only be less strong. The fact that ET estimates do not improve with improvements in soil moisture is likely because ET is not so sensitive to changes in soil parameters, even for different climate conditions. This is for instance shown in a paper I wrote in 2009 (Teuling et al., 2009), in which I investigated effects of soil parameters on soil moisture and ET. In short, effects of soil parameters on soil moisture are generally large, but effects on ET are small. This is primarily caused by the main effect of soil parameters which is a shift in the mean, rather than*

*the dynamics (this is also consistent with the strong contribution of bias to MSD as reported by the authors). This should be discussed better.*

ANSWER:
We thank the Referee for providing this very important comment. The revised manuscript will focus mainly by addressing more specifically the fact that changes in soil parameters limited the improvement to soil moisture causing little effect on on evapotranspiration.

*COMMENT:*
*Related to this is the question why rooting depth was not optimised along with the soil parameters. If the model rooting depth does not reflect the actual root profile, optimization of soil parameters along will not lead to a better estimation of available soil water. These choices should be better motivated.*

ANSWER:
We used site specific data in order to prescribe rooting depth for the analysed sites (as mentioned in original manuscript Page 7, line 32 – Page 8, line 2). The decision was made to avoid prescribing more than one rooting depth parameter for individual sites given that rooting depth in JULES is defined per Plant Functional Type. This is indeed a limitation in our analysis and will be mentioned in our revised manuscript.

*COMMENT:*
*In making a case for the validity of their research question, the authors miss out on an important body of literature on spatio-temporal characteristics of soil moisture fields. It has been shown by numerous studies that while soil moisture generally shows a large spatial variability, individual points maintain their rank while the mean changes. This insight started with the "classical" Vachaud et al. study, but numerous other studies (for instance Teuling et al., 2006, Mittelbach and Seneviratne, 2012, among others) have reported similar behaviour. This behaviour implies a relatively small spatial variability of fluxes, which was explained from a theoretical perspective by Albertson and Montaldo (2003) and explored Teuling and Troch (2005), among others. In effect, based on these studies, the hypothesis could also be formulated more neutral in the form of a null-hypothesis: "A reduced scale mismatch does not lead to LSM flux estimates closer to eddy covariance observations..". The results could subsequently be interpreted as insufficient evidence to reject this hypothesis. In any case the introduction should be changed to include a discussion on the relation between point-scale and large-scale soil moisture. This can then also be used to explain the reference to Franz et al. (2012), which now conflicts with the hypothesis (if CRNS and in-situ soil moisture have already been reported to be similar, how can the authors still expect to find a difference in fluxes?). These comments do not make the research any less relevant, but the phrasing of the hypothesis should be in line with the "state-of-the-art", and not a convenient selection thereof.*

ANSWER:
We thank the Referee for this comment. We will expand the discussion and include additional literature as suggested by the reviewer. Based on the reviewer's suggestion, we will also revise our hypothesis accordingly. In addition, please notice that Franz et al. (2012) compared a network of multiple point-scale sensors within a single CRNS. This network consists of multiple profiles of soil moisture estimated by point-scale sensors within the same CRNS footprint.

*COMMENT:*
*In applying CRNS and PS soil moisture products, the authors only consider the shallower PS soil moisture observation in order to comply with the CRNS observation depth. While this makes sense given the goal of the study (scale mismatch and not a comparison of observation techniques), it is not*

*sufficiently recognized that this introduces an unfair disadvantage to the PS data. There might be important information in deeper PS observations, in particular during soil moisture-limiting conditions, that is not being considered in this study. It is thus crucial to make a clear distinction between the scale aspect and the observation technique, and acknowledge in the discussion in the discussion that PS soil moisture might give better results when all available observations are used. A better alternative would be to redo the analysis using all observation depths in addition to using only the shallow observations, but this would likely require a substantial amount of work.*

ANSWER:
We thank the reviewer for this comment. We agree that not using the deeper soil moisture observation can introduce disadvantages to the point-scale measurement in the comparison. Notice, however, that most Ameriflux sites provided only data from shallow sensors with exception of WR (down to 50 cm) and MO (down to 100 cm), respectively. Understanding the impact of deeper soil moisture dynamics is indeed very interesting to be investigated but it is beyond the original scope of our study which focus on understanding more directly the impact of horizontal footprint. We will include a statement to better clarify this in our revised manuscript.

REFERENCES
Franz et al, 2012. Field validation of a cosmic-ray neutron sensor using a distributed sensor network. Vadose Zone Journal 11 (4), doi:10.2136/vzj2012.0046.

Teuling et al, 2009. Parameter sensitivity in LSMs: An analysis using stochastic soil moisture models and ELDAS soil parameters. J. Hydrometeorol., 10(3), 751-765.

---

## Author Comment (AC3) · 10 Jan 2017

Answer to interactive comment by Anonymous Referee #3

We would like to thank the Referee for carefully reviewing the manuscript. We thank the Referee for his comments on the structure of our manuscript, specification and quantifications, and the English language. We will address these issues in the revised version of the manuscript. Below, we focus on the comments that concern with methods chosen and the conclusions drawn.

*GENERAL COMMENTS*

*COMMENT:*

*The manuscript is quite well written and clear, even though some parts should be reduced and summarized. The topic is surely of interest for the HESS readership as cosmic-ray probes represent a new technology for ground measuring soil moisture over large areas. Therefore, we need to assess the impact of this new technology for improving land surface modelling. The paper describes the calibration of JULES land surface model with PS and CRNS at different sites in US. The manuscript is well conceived and applied over a large number of sites thus obtaining reliable and robust results. However, I mostly agree with the comments of previous reviewers, and particularly I believe that several aspects should be improved/changed before the publication. I reported below a list of the general comments to be addressed with also the specification of their relevance.*

ANSWER:
We thank the referee for his positive comments regarding the relevance of our work to the HESS community.

*COMMENT:*
*1) MAJOR: I found some of the explanations/justification of the results given in the paper quite weak. They appear to me as speculations, not supported by the performed analyses and results. For instance, I refer to:*
*(A) The comparison between PS and CRNS soil moisture data (section 3.1.3) shows that soil moisture timeseries are quite similar. The authors expected better performances at homogeneous sites but it was not the case. As shown in "temporal stability" papers (see also Teuling report), PS measurements are usually very well correlated with large scale measurements. Therefore, I expect good correlations. In my opinion, the good (bad) performances are mostly related to the good (bad) quality of soil moisture observations that may be affected by a number of factors (e.g., soil texture, sensor malfunctioning, ...). Therefore, theoretically I could expect that CRNS are better than PS measurements, but due to measurement uncertainties and errors, the larger support scale of CRNS is masked out by the (likely) lower quality of their measurements.*
*This important aspect, i.e., the quality of soil moisture observations, should be carefully addressed in the paper.*

ANSWER:
We thank the Referee for this valuable comment. As also suggested by Referee #2, we will include a discussion on temporal stability/soil moisture scale issues in the revised version of the manuscript. We will also discuss the impact of quality of observations on the soil moisture signal in the revised version of the manuscript.

*COMMENT:*
*(B) The authors attributed the low differences in estimating surface energy fluxes when PS and CRNS measurements are considered to the weak coupling in JULES between soil moisture and evapotranspiration. Actually, I do not believe it is the case, but it should not be a speculation. It should be tested. If the authors want to give this message, they should demonstrate that with a*

*different model or land surface scheme the differences are higher, and likely CRNS is better that PS data (as expected at the beginning). Therefore, I suggest changing the conclusions or, better, implementing an additional LSM and demonstrate the results through a scientifically sound approach.*

ANSWER:
We thank the Referee for this comment. This is a point that was also raised by the other two referees. We agree that our use of the word coupling was not appropriate as also pointed out by the other reviewers. Our interpretation is, in fact, in line with Reviewer #2 comments which should emphasize that the results obtained in our study are a consequence of calibrating soil parameters, rather than the soil moisture – evapotranspiration coupling in JULES. This will be appropriately revised in the next version of the manuscript.

*COMMENT:*
*(C) The range of reasons reported at page 13, lines 13-34 are only speculations. I suggest removing.*

ANSWER:
We thank the referee for his/her comments. Note, this issue was also raised by Referee #1. We will re-assess this section of the manuscript for the revised version of the manuscript.

*COMMENT:*
*2) MAJOR: I found quite strange that by using the default parameter values performs the same than using the parameter values calibrated on soil moisture data in terms of evaporation fraction (EF) estimation. Even though soil moisture data were of low quality, or the coupling between soil moisture and EF is weak, soil moisture observations represent local data that should give some information to the model. Therefore, I expected better results with respect to the default parameterization. What happens if JULES is calibrated on EF data? How the corresponding modelled soil moisture data compare with PS and CRNS observations? By looking at the results reported in the paper, it seems that using soil moisture observations is needless if we have the purpose of improving land surface modelling. I suggest the authors to improve the discussion and the analysis of the results.*

ANSWER:
This is an important point raised by the reviewer and it is in line with Referee #2's comments. Our interpretation is, in fact, that the results obtained in our study are a consequence of calibrating soil parameters, rather than the soil moisture – evapotranspiration coupling in JULES. We will improve the discussion and analysis for the revised version of the manuscript as also recommended by the reviewer.

*Specific comments*

*COMMENT:*
*P4, L32: The gap-filling of 30 days seems to me a very large window. Does it affect the results? Some tests should be made.*

ANSWER:
We thank the reviewer for raising this point. This was also mentioned by reviewer #1. We provide a table below summarising the average (average of the seven forcing variables) percent gap for each site. Precipitation was not gap-filled; missing points were set to zero instead. Overall, average gaps vary among sites between near zero to 15%. As pointed out by the reviewer, this can introduce

some uncertainty in the analysis and we will highlight data when describing the dataset used in this study. We will also mentioned in the text the original gap period in the data.

| Site | Percentage missing hours filled: mean (range) | Size of time series in years |
|------|----------------------------------------------|------------------------------|
| UM | 7 (2-12) | 1.5 |
| DC | 1 (0-5) | 3.7 |
| SO | 7 (0-15) | 3.8 |
| KE | 1 (0-3) | 4.6 |
| ME | 0 (0-0.1) | 1.6 |
| SR | 2 (0-14) | 3.6 |
| CS | 2 (0-5) | 3.7 |
| MM | 1 (0-2) | 2.7 |
| TR | 0.1 (0-0.2) | 3.6 |
| AR | 10 (8-14) | 2.5 |
| WR | 2 (0-4) | 2.6 |
| MO | 3 (1-7) | 2.7 |

*COMMENT:*
*P10, L18-22: The difference in sensing depth might be the cause of some of the differences between PS and CRNS. However, it could be checked with specific analysis. Otherwise, I suggest removing.*

ANSWER:
We will remove this text.

*COMMENT:*
*P12, L3-6: It is not clear to me what the authors want to demonstrate with this analysis. Please clarify.*

ANSWER:
With this analysis we wanted to see if surface energy partitioning estimation did improve during periods of soil moisture stress, when soil moisture was below the critical point. During these periods soil moisture provides a first order control on evapotranspiration. We will improve this discussion accordingly in the revised manuscript, as also suggested by Referee #1.

*COMMENT:*
*P16, L4-11: As mentioned above, I found not scientifically sound to attribute the low performances in term of EF improving to the weak coupling of JULES. Moreover, it's not clear to me the discussion of the value of absolute soil moisture with respect to anomalies.*

ANSWER:
We thank the reviewer once again for making this excellent point. As we previously mentioned, our use of the word coupling was not appropriate in the manuscript. Our interpretation is, in fact, in line with Reviewer #2 comments which should emphasize that the results obtained in our study are a consequence of calibrating soil parameters, rather than the soil moisture – evapotranspiration coupling in JULES. This will be appropriately revised in the next version of the manuscript.

---

## Author Response (AR1)

**Final answer to review comments on manuscript originally entitled "Reducing soil moisture measurement scale mismatch to improve surface energy flux estimation" by Iwema et al. submitted to Hydrology and Earth System Sciences**

We thank the Editor for his work and guidance. We address the comments of each referee below. At the end of this answer a marked up version of the revised manuscript is included. For clarity we first address the main issues mentioned by the Editor, based on comments by the three Referees:

1. Consider restructuring the manuscript and improving parts of the manuscript like the discussion of the literature (incomplete as detailed by reviewer #2).

- We have moved Section 2.3 "Soil moisture data comparison methodology" forward; now it is Section 2.2.
- Figure 1, Figure 8a, and Figure 13 (numbering as in the original manuscript) were removed.
  Figure 3 was moved from the main part of the manuscript to the newly created supplemental material, more precisely Supplement 1.
- We now use the Root Mean Squared Error (RMSE) of latent heat flux as principle validation metric. We have moved Figure 8b, presenting the RMSE of the evaporative fraction, to Supplement 2. We now mention the evaporative fraction analysis only briefly in Section 3.3 of the new manuscript. We produced a figure equivalent to Figure 8b, showing the relative improvement in latent heat flux RMSE against the relative improvement in soil moisture RMSE to the main part of the manuscript. This provides a more consistent structure. We have rephrased sentences in Section 3.3 "Validation", now 3.3 "Validation of the single-objective calibration against eddy-covariance observations", in accordance with this change.
- We have moved Section 3.4 "Were calibrated parameter values physically feasible?" to the new Supplement 3. This allowed us to present a more coherent story without any unnecessary distractions. We refer to this supplement from Appendix 2, which itself was Section 3.6 in the old manuscript ("To which parameters were soil moisture and latent heat flux most sensitive?").
- We have completely rephrased Section 3.5 (now 3.4) "Two-objective calibration against soil moisture and latent heat flux" to provide a clearer interpretation of the results, after comments by Referee #1.
- We have removed Section 3.7 "Could JULES model structure explain the limited improvement in surface energy flux estimation?" and Section 3.8 "Discussion", after comments by Referee #1. We have allocated the paragraphs from these sections to Section 3.1, 3.2, 3.3, and 3.4. This has improved the structure and readability of our manuscript.

**2. The link between results and conclusions is questioned by the reviewers. In particular, the weak coupling between ET and soil moisture is not so obvious from the results.**

- We have revised our conclusions after comments by all three referees. We don't conclude any longer that JULES has weak soil moisture – evapotranspiration coupling. We attribute the lack of differences between calibrating against the two different soil moisture observations products (PS and CRNS) to (1) the limited effect of calibrating soil parameters on evapotranspiration, (2) the implications of spatio-temporal stability theory, (3) data quality properties of the soil moisture observation techniques, and (4) the self-adjusting behaviour of the wilting point and critical point soil moisture parameters after calibrations.

3. Some topics require additional attention: the role of deeper soil moisture and the rooting depth, and also the quality of the different types of soil moisture measurements.

- We have addressed the issues of the value of deeper soil moisture and rooting depth in our answer to the comments by Referee #2. We have included discussions on these issues in our "Results and Discussion" section.
- We have addressed the quality of different types of soil moisture measurements (uncertainties in Cosmic-Ray Neutron Sensor and point-scale data), as raised by Referee #3, in our "Results and Discussion" section.

In addition to these major points, we have improved the phrasing in some sentences not mentioned by the referees. We have removed a few paragraphs (detailed in the answers to specific comments by the referees) to decrease the size of our manuscript and to improve its readability. We have changed one reference, to Gupta et al. (1998), which was on page 13, line 38 of the original manuscript, to Gupta et al. (1999). Two other references were found missing in the literature list and were therefore added (Gupta et al., 2009; Finkelstein and Sims, 2001).

Finally, we have changed both the title and hypothesis after comments by Referee #2 Ryan Teuling. The title is now:

"Land surface model performance using cosmic-ray and point scale soil moisture measurements for calibration"

The new hypothesis is:

"Reduced scale mismatch does not lead to LSM flux estimates closer to eddy covariance observations"

Sincerely,

Joost Iwema and co-authors

**Answer to interactive comment by Anonymous Referee #1**

We would like to thank the Referee for carefully reviewing the manuscript. A point-by-point reply to the comments is provided below with original comments shown in Italic font.

**OVERVIEW**

The presented manuscript investigates the potential impact of the measurement scale for calibration of a land surface model. For this purpose, observed and simulated land surface data at 12 sites on the continental US from several sources including Ameriflux, COSMOS and NLDAS was used. Point scale soil moisture data was compared to cosmic ray soil moisture retrievals. Furthermore, land surface simulations at the nine sites were done on an individual basis using JULES. At each sites, JULES was calibrated with cosmic ray data, point scale soil moisture data and eddy flux measurements. Model results were evaluated with eddy flux and soil moisture measurements. The case study demonstrates the added value of cosmic ray measurements at the model scale compared to local scale soil moisture measurements.

However, the study needs a major revision that addresses readability in the following: Reduce/clarify abbreviations, restructure part of the manuscript, improve English / sentence structure, remove speculations, be more specific /quantitative at a number of instances. There seems to be an issue with the data presented in Figure 7 concerning site MO.

The figures require further formatting. I suggest reducing the number of Figures. This allows the reader to focus on the essential messages of the study. I disagree with the outcome that coupling of soil moisture and latent heat flux is weak in JULES (e.g. see comment to Figure 9). Further suggestions in the Specific comments.

**ANSWER:**

We thank the referee for this evaluation of the manuscript. We have addressed the issues mentioned above by the referee to increase the readability. This issue was also raised by the other two referees. The issue with Figure 7 is explained in the 'specific comments'.

All three Referees disagreed with our conclusion that the coupling between soil moisture and latent heat flux in LSM JULES is weak. We agree that our use of the word 'coupling' was not appropriate as also pointed out by the other referees. Our interpretation is that the results obtained in our study are mainly a consequence of calibrating soil parameters, rather than the soil moisture – evapotranspiration coupling in JULES. Other reasons playing a role are the implications of the spatio-temporal stability of soil moisture measurements, the quality of the soil moisture data, and the self-adjusting behaviour of the wilting point and critical point soil moisture.

**GENERAL COMMENTS:**

**COMMENT:** The paper exhibits a clear novelty by quantifying the impact of using cosmic ray soil moisture data for calibration as compared to local point soil moisture measurements. The study fits the scope of the HESS journal and deserves to be published in HESS after major revision.

The conclusions reached in the manuscript are not clear enough. I also found different conclusions from the data and results presented. For details: See Specific comments to Chapter 4. The scientific methods and assumptions were well chosen and represent state of the art.

**ANSWER:** We thank the reviewer for his/her positive comments regarding our manuscript and its relevance to the HESS community. We have addressed the issues regarding the clarity and inconsistency of our conclusions in the revised version of the manuscript as pointed out in detail by the reviewer's comments.

**COMMENT:** Description of experiments and calculations need to be revised. I suggest following new structure: Chapter 2.1 can remain there. Then, Chapter 2.3 should be changed to Chapter 2.2 as soil moisture data should be compared before calibration or modeling. Then explain JULES, then JULES forcing and initial conditions, the following Chapters can remain in place.

ANSWER: We have adopted the structure suggested by the referee.

**COMMENT:** The results chapter needs a new structure. The results are presented in the right order but intermittent by discussions that are out of place because there IS a Chapter "3.8 Discussion". A more clear structure would be either consistently "3. Results and Discussion" or "3. Results; 4. Discussion; 5. Conclusion". Please, stick to either one but do not mix. The topic is complex and in general well addressed, but a new structure will increase readability and will make writing the paper more easy.

**ANSWER:** We agree with the referee's comment. We have created a section "3. Results and Discussion". Section 3.7 and Section 3.8 were removed from the revised manuscript. The different paragraphs were assigned to Section 3.2, 3.3, and 3.4 of the revised manuscript.

**COMMENT:** The title reflects the content. However, I would suggest a modification of the title to e.g. "Improved land surface processes by calibration with cosmic ray soil moisture measurements at the model scale".

**ANSWER:** We thank the reviewer for making this point, which was also raised by Referee #2. We have adopted the title proposed by Referee #2 Dr. Teuling: "Land surface model performance using cosmic-ray and point scale soil moisture data for calibration".

**COMMENT:** The abstract is concise and summarizes the paper well. It may be modified if conclusions are changed.

**ANSWER:** We appreciate that the Referee found our abstract to be concise and a good summary of our manuscript.

**COMMENT:** In general, there is a large number of abbreviations (e.g. PS), symbols (in formulas), short names (e.g. smcrit). This makes the paper very difficult to follow. It is necessary to use either abbreviations, or symbols also in the text, ommit short names and in general write the names out more often. This paper almost needs a List of Abbreviations. Please, reduce them.

**ANSWER:** This is a valid point raised by the referee. In the revised version of the manuscript we have reduced the number of abbreviations. We have however kept certain abbreviations that are commonly known in the land surface modelling community and in the soil moisture measurement community. Another reason to keep these abbreviations is that writing them out would increase the

size of the manuscript considerably and labels in figures would not fit with a readable font. These include "PS" (point-scale), "CRNS" (Cosmic-Ray Neutron Sensor), "LSM" (Land Surface Model), "SM" (soil moisture), and "JULES" (Joint UK Land Environment Simulator). We have removed the abbreviations "EC" (eddy-covariance) and "OF" (objective function).

**SPECIFIC COMMENTS**

**COMMENT:**

Page 1 line...: 9: "sometimes" is too unprecise. Rephrase e.g. "can be calibrated" 10: EC, PS, LSM, CRNS, JULES - How can the abbreviations in the abstract be reduced? 12: Is there a term "soilevapotranspiration"? I suggest "soil and evapotranspiration" or "soil-evaporation". 17: "CRNS calibrations" – What is actually meant is "LSM calibration" or something alike. 30: "atmospheric circulation" - In the UK? Be more specific here or do not mention it. 30: "Because", is "because" naturally a start of the sentence?

**ANSWER:**

Page 1 line 9: We have replaced "sometimes" with "can be calibrated".

Line 10: We have removed all abbreviations except "JULES" and "USA" from the abstract. We kept "JULES" because this is how the model is generally referred to by the land surface modelling community. We assume the abbreviation "USA" is commonly understood.

Line 12: The term "soil-evapotranspiration" has been replaced with "soil and evapotranspiration". Line 17: We have changed "We found that simulated surface energy partitioning did not differ substantially between the PS and CRNS calibrations" to "We found that simulated surface energy partitioning did not differ substantially between both calibration strategies".

Line 30: The paragraph on line 28 of page 1 to line 2 of page 2 has been changed to: "Land Surface Models (LSMs) solve the surface mass (including water), energy, and momentum balances to provide the weather and climate prediction models with lower boundary conditions. The land surface has been shown to play an important role in global atmospheric circulation (Koster et al., 2004). Because the soil moisture state and surface fluxes are so closely connected, it is important to accurately simulate these simultaneously (Henderson-Sellers et al., 1996; Richter et al., 2004; Seneviratne et al., 2010; Dirmeyer, 2011; Dirmeyer et al., 2013)."

**COMMENT:**

Page 2 line...: 1: "believed" - It is for THESE authors important or it is not. Maybe, why is it important. There is no "believe" in science. 4: "processes." Reference missing. 4: Remove "however". no added value. 14-15. Unclear, specify or rephrase. 22: "in-situ soil moisture" – I suggest to be consistent throughout the paper.in-situ or point scale is the same. So I suggest using one phrasing only. 24: soil moisture "IS" spatially and add reference e.g. Qu et al. 2015 in GRL. 26-28: "EC footprint average soil moisture" or "LSM grid cell" average soil moisture depends on the objective. As is, it is confusing. Rephrase to be clear. 31: Is this the research question? There is another research question on the next page. Either both should be allocated together or only one objective / research question is needed. As is, it is confusing.

**ANSWER:**

Page 2 line 1, please see answer to previous comment (page 1 line 30). Line 4: The references in the next sentence (line 4-5) have been added to this sentence and the sentence has been changed to reduce the size of the manuscript (Lines 3-5 of the revised manuscript): "The increasing complexity of land surface models over the past decades (Sellers, 1997; Seneviratne et al., 2010) has brought with it an increasing number of parameters, with values not easily defined with in-situ measurements because of scale mismatch"

Line 14-15: We rephrased to:

"In an effort to develop global hydrometeorological monitoring and prediction capabilities (Wood et al., 2011), hydrological models and LSMs are now increasingly being applied at the finer `hyper-resolution scale' with grid cells of about 1 km2."

Line 22: We respectfully disagree with the Referee on this point, because Point Scale soil moisture sensors, Cosmic-Ray Neutron Sensor soil moisture sensors, and Eddy-Covariance sensors are all insitu; they are on site. We use Point Scale soil moisture sensors to refer to those placed within the soil, and which represent a footprint of a  $\sim$ 4 dm3 only.

Line 24: "Soil moisture can be spatially non-uniform" has been changed to: "Soil moisture is spatially non-uniform" and the suggested reference will be added.

Lines 26-28 Have been rephrased to (lines 21-22 of the revised manuscript): "Therefore, soil moisture measurements best (i.e. most effectively) representing the soil below the eddy-covariance tower's footprint should be used when the performance of a land surface model is evaluated". Line 31: The question phrased here is our research question. On page 3, lines 22-23 of the original manuscript we phrased a hypothesis for our research question. We have rephrased this text to: "In order to answer our research question (page 3, line 8 of the revised manuscript) we hypothesise that *reduced scale mismatch does not lead to LSM flux estimates closer to eddy covariance observations.*" Please note that we have changed our hypothesis to a null-hypothesis as proposed by Referee #2. We have moved the hypothesis to immediately after the research question.

**COMMENT:**

Page 3 line...: 5 "usually assumed" - this is rather a fact due to soil heterogeneity. 10: One sentence paragraphs should be avoided. Also the use of "which" as often as it is used in the present manuscript, should be avoided. In English, short sentences are much better understood and much clearer. One sentence is preferably 1-2 lines only. 12: add the German CRNS network (Baatz et al 2015) 13: Repetition. 16: "similar" - be more specific. Similar is least informative and makes room for interpretation. 17: which – split the here sentence. 21: which - rephrase to "These sites" also look up the difference between which and that. Is the information you give with "which" really necessary? Then put it in one single sentence as it is worth a single sentence. 27: that – could that be removed here? 30: "Before our modelling exercise" ... This sentence should be moved up within the paragraph. First things first. 30-33: A shorter sentences are desirable.. It can be two or more sentences.

**ANSWER:**

Page 3, line 5: Due to comments by Referee #2 and Referee #3, we have kept this sentence as it is, but is was moved up (page 2, lines 24-27), to before the research question. The other two referees argued that based on temporal stability theory more soil moisture observation points in space do not necessarily yield better surface energy fluxes.

Line 10: This one sentence paragraph has been combined with the following paragraph.

Line 12: The reference to Baatz et al. (2015) has been added.

Line 13: This sentence has been removed and the information has been added to the first sentence of this paragraph instead: "The CRNS (Zreda et al., 2008) is an above-ground passive sensor which utilises natural cosmic-ray neutron radiation to estimate soil moisture content in the top 10-70 cm. The sensor's footprint area has a radius of about 100 to 300 m surrounding the above-ground sensor (Figure 2; Desilets and Zreda, 2013; Kohli et al., 2015)."

Line 16: Changed to: "Franz et al. (2012) showed soil moisture estimated from CRNS neutron counts differed less than 20% from the average of a co-located point scale soil moisture sensor network at a site in Arizona."

Line 17: We have rephrased this sentence to: "Unlike wireless point scale sensor networks, both the GPS and CRNS technology provide an integrated soil moisture measurement over the entire support volume (Larson et al., 2008; Zreda et al., 2008)."

Line 21: We have changed to: "Twelve of these sites provided sufficient ..."

Line 26-27: We have rephrased to: "We did this by calibrating parameters of the Joint UK Land Environment Simulator (JULES; Best et al., 2011) against Point scale and Cosmic-Ray Neutron data separately."

Line 30-33: We have moved the sentences on lines 30-33 to the beginning of the paragraph, lines 31-33 of the revised manuscript.

**COMMENT:**

Page 4 line...: 10: "used data" - Specify "data". 10: remove brackets and specify e.g. "the upper first 30 cm". 15: Remove "Similar". 18: "CRNS integrated soil moisture" ... rephrase for better understanding e.g. soil moisture integrated over depth from CRNS soil moisture, hereafter referred to as CRNS soil moisture retrieval. 23: Split sentence, remove while. 31: remove which, split sentence 32: "More than 31 days were gap filled" using average diurnal pattern. This sounds like a really high uncertainty. Is this the case for precipitation, too? Is the high uncertainty reflected in the results? If so, where? Is it feasible to mark this in the Figures? Is it feasible to remove these periods from the calibration period? How much of modeled periods was filled with diurnal patterns?

**ANSWER:**

Page 4 line 10: We have specified "data" as follows: "We used point scale soil moisture data from the soil layers up to 30 cm depth only for consistency among all sites. Our main objective was to investigate the difference in information content due to two soil moisture measurement techniques' different horizontal scales in relation to the eddy-covariance footprint, rather than to compare the measurement techniques themselves."

Line 15: "Similar" has been removed.

Line 18: We moved this sentence to the Section "Soil moisture data comparison methodology" and rephrased the last lines of that section to: "We compared PS soil moisture with CRNS soil moisture values computed from vertically homogeneous soil moisture values obtained from the observed neutron counts using the COsmic-ray Soil Moisture Interaction Code (COSMIC; Shuttleworth et al., 2013)."

Line 23: This sentence was split and the second part was rephrased to: "We used data version 3.4 (Goulden et al., 2015) for the three California Climate Gradients sites."

Line 31: We have split the sentence and removed "while" as suggested by the reviewer.

Line 32: We thank the reviewer for raising this point. This was also mentioned by reviewer #3. We provide a table below summarising the average (average of the seven forcing variables; Table a) percent gap for each site filled with this method and a figure (Figure a) showing how large the gaps were. This figure shows the percentage missing hours for each gap size category. If the bars for one variable at one site would be summed, this would yield the total percentage of missing hours. Overall, average gaps vary among sites between near zero to 15%. Gaps larger than 15 days occurred at:

- UM: downward shortwave and downward longwave radiation (9% of time)
- DC: net radiation (4% of time)
- KE: downward longwave radiation (4% of time)
- SR: downward longwave radiation (8% of time)
- CS: downward shortwave radiation (3% of time)
- MM: pressure (2% of time)
- AR: temperature, wind speed, and relative humidity (2-4% of time)
- WR: precipitation and pressure (3% of time)

**- MO: precipitation (2% of time)**

Summarising, large gaps (over 15 days) occurred for no more than 10% of time at any site and occurred at most sites for longwave and shortwave radiation only. Precipitation is likely to be the primary gap factor affecting soil moisture but its occurrence was minimal. Nevertheless, we noted this issue in the manuscript (Section 3.3 of the revised manuscript, see below).

In this study, we used the most regular freely-available data sets to the best of our knowledge to ensure reproducibility. We noticed that outcomes for sites with few large data gaps did overall not differ from the outcomes on sites with more large data gaps. Therefore, we argue that although the data gaps may introduce some uncertainties in the model results, our major conclusions remain unchanged. We discuss the most important large data gaps in Section 3.3:

"A month long gap in longwave and shortwave downward model forcing data occurred in April 2012. This might have affected the results for this site."

**and in Section 3.4 of the revised manuscript:**

"An issue that could have had effect on our results is the occurrence of gaps in the model forcing data. Gaps in observed meteorological data are often inevitable and must be filled for a land surface model to be run. Percentages of missing hours differed between 0% and 15% at our sites. Gaps larger than fifteen days filled with the moving window gap-filling procedure occurred mainly for downward shortwave and/or downward longwave radiation; at sites UM, KE, SR, DC, TR, and CS. At site WR a data gap of 29 days in precipitation and air pressure occurred. Especially the gap in precipitation data can have negatively affected the model results at this site."

Please note that the gap filling script was actually designed to fill gaps up to 30 days and not 31; we have changed this detail in the new version of the manuscript (page 5, line 24). Precipitation was not gap-filled; missing points were set to zero instead. Finally, we found that Table A1.2 contained errors; we have corrected this in the new version of the manuscript (page 37).

| Site | Percentage missing hours filled: | Size of time series in years |
|------|----------------------------------|------------------------------|
|      | mean (range)                     |                              |
| UM   | 7 (2-12)                         | 1.5                          |
| DC   | 1 (0-5)                          | 3.7                          |
| SO   | 7 (0-15)                         | 3.8                          |
| KE   | 1 (0-3)                          | 4.6                          |
| ME   | 0 (0-0.1)                        | 1.6                          |
| SR   | 2 (0-14)                         | 3.6                          |
| CS   | 2 (0-5)                          | 3.7                          |
| MM   | 1 (0-2)                          | 2.7                          |
| TR   | 0.1 (0-0.2)                      | 3.6                          |
| AR   | 10 (8-14)                        | 2.5                          |
| WR   | 2 (0-4)                          | 2.6                          |
| MO   | 3 (1-7)                          | 2.7                          |

Table a: Percentage of missing hourly model forcing records. In the middle column the mean over all input variables is shown along with the range over all input variables. In the right column the length of the simulation period is shown.

Figure a: Missing hours as percentage of the total hours in the calibration/validation periods. The statistics are shown per forcing variable. Sw is downward shortwave radiation, Lw is downward longwave radiation, Rain is precipitation, T is air temperature, W is wind speed, RH is relative humidity, and P is atmospheric pressure. Please note that for convenience the label Lw is also used for site DC; instead of net radiation. The four categories split the missing hours between gaps of less than 3 hours, of between 3 hours and 7 days, of between 7 days and 15 days, and gaps larger than 15 days.

**COMMENT:**

Page 7 line...: 8: Rather "Calibration Approaches". Where is the "two-objective calibration" in the methods section? 10-12: Sentence too long. Rephrase into at least two sentences. 12: Add reference (Shuttleworth et al . 2013) 14-15: This reads difficult. Either write saturated hydraulic conductivity instead of sathh or use the symbol introduced in Eq. 5. Same for all other symbols throughout the manuscript. 22-25: Shorter sentences. 27: This this.... rephrase

**ANSWER:**

Page 7 line 8: Title has been changed to "Calibration and validation approaches". We have combined this section with the next section ("Validation approach") and now introduce the "two-objective calibration" in this section.

Lines 10-12: The sentence has been rephrased to: "We chose to calibrate simulated neutron counts against CRNS observed neutron counts using COSMIC (Shuttleworth et al., 2013) to translate simulated soil moisture profiles into equivalent neutron counts."

Please note that the reference to Shuttleworth et al. (2013) was already provided in Section 2.3 of the old manuscript (Section 2.2 in the new manuscript).

Lines 14-15: We have rephrased: "These included Mualem-Van Genuchten shape parameter b, the water entry pressure parameter (sathh), and the saturation hydraulic conductivity ( $K_{sat}$ ). The critical point ( $\theta_{crit}$ ) and wilting point ( $\theta_{wilt}$ ) soil moisture content parameters from the evapotranspiration limitation factor were also calibrated."

Lines 22-25: Sentence has been split as follows: "They were constructed by computing the minimum and maximum parameter values for the entire soil texture triangle (based on Wösten PTF). Three organic matter contents were taken into consideration, yielding three triangles. Cay percentages above 70% were excluded to avoid extreme values for parameter b especially."

Line 27: The word "thus" has been removed: This yielded different critical soil moisture content ranges in terms of soil moisture content (m3m-3) for different sites.

**COMMENT:**

Page 12 line ...: 32: Remove "It should be noted however that".

**ANSWER:**

Page 12 line 32: This sentence has been changed to: "Especially for the saturation hydraulic conductivity, the parameter calibration range was non-linear, with a factor 1000 difference between the upper and lower boundaries."

This entire section has been moved to the newly created supplemental material, Supplement 3.

**COMMENT:**

Page 13 line...: 6: I suggest to put "and also containing stones" to the end of sentence or another place and remove brackets. 7: Here you write sathh out. Much better. However, the sentence may be moved to the discussion section. 13: I suggest to discuss these reasons. Actually, it is worth investigating each of the points to either accept them or rule them out. Mentioning all of these points / reasons is not getting the manuscript closer to the objective. 35: The "two-objective calibration" was not mentioned before. However, it is good to have it.

**ANSWER:**

Page 13 line 6: Sentence has been changed to: "At the KE site, the soil was reported to be coarser than actually reported by the HWSD and to contain stones."

Line 7: We have renamed the Discussion section and rearranged sections to increase the readability. Therefore we could keep this sentence within the section it was before.

Line 13: This issue was also raised by Referee #3. We have decided to move the entire section to the supplemental material, Supplement 3. We refer to it from Appendix 2, which now contains what was previously Section 3.6 "To which parameters were soil moisture and latent heat flux most sensitive?" The reason is this section presents more a quality check of the calibration results than it is a contribution to answering our research question, as mentioned by Referee. We now discuss the most plausible reasons only. We quote the new text here:

"Several reasons have been proposed why apparently non-physically realistic parameter values were found, of which we discuss the most plausible ones here. Richards' Equation applicability at larger horizontal scales is questionable for multiple reasons (Beven and Germann, 2013). The foremost reason is hydraulic continuity, upon which Richards' Equation is based, does usually not occur in the field over distances of multiple metres. This is due to horizontal and vertical soil heterogeneity (e.g. macro-pores, soil layering in organic matter and particle sizes, stones) not (properly) accounted for in the model. The form of Richards' Equation implemented in JULES does not take these natural features into account and therefore effective parameters can be expected to differ from values representing a soil with a homogeneous matrix only. That means the process represented by the equation is not the same as the processes occurring in the field soil. Therefore the parameters' roles in JULES differ from their theoretically assumed roles, and so different effective parameter values can be expected. Explicitly taking into account heterogeneity issues like macro-pores could improve JULES' performance (Rahman and Rosolem, 2017), but doing so was beyond the scope of our study. The vertical discretisation of the soil (layers of 10, 25, 65, and 200 cm) may not be suitable for solving Richards' Equation. Layer thicknesses of no more than one centimetre near the surface are required to accurately simulate water fluxes. Insufficiently fine discretisation at both field (~eddy-covariance footprint) and hydrological catchment scale may not yield realistic water fluxes using realistic parameter values. (Smirnova et al., 1997; Lee and Abriola, 1999; Van Dam and Feddes, 2000; Downer and Ogden, 2004; Beven and Germann, 2013). Testing the effects of this issue on our results, which would entail changing the vertical discretisation of the soil column in JULES, was beyond the scope of our study. Finally, due to the simplifications of natural processes inherent to models, physically realistic parameters often provide unrealistic results (Gupta et al., 1998)."

Line 35: In the new version of the manuscript we have introduced the two-objective calibration in Chapter 2 Data and Methods instead.

**COMMENT:**

Page 14: line...: 5: comma before respectively. 7: was instead of "could have been" 7: "similar" quantify or remove. 11: Suggestion: Move "to have obtained such automatic improvement" to the end of sentence 16: "improved soil moisture and latent heat flux". If calibration is done for both, both should be improved, no? 18: What does EF stand for? Too many abbreviations. Also WO: : :etc. There should be a way to distinguish places/sites from variables from acronyms. 19: "coincidence" – Not really, is it? 20: "Similar" - be precise. 21: "this" means what? context not totally clear. Rephrase. 27: "Generally quite weak" - Reference in literature? I cannot quite follow here. 28: What is the reasons that at these sites strong coupling is expected? At what time? Is the calibration done during the time of strong coupling? Very unclear how this conclusion is reached. I suggest an individual point of discussion. This is also the point where I cannot follow the conclusion drawn. It would be a major setback of JULES which needs to be justified much stronger. Above all, the results in the Figures show a meaningful difference in ET due to calibration / calibration method.

**ANSWER:**

Page 14 line 5, line 7, line 11: Section 3.5 (now Section 3.4) has been completely rephrased. These errors were therefore removed.

Line 16: Please note that in two-objective calibration simultaneous improvement in both is not necessarily expected (Gupta et al., 1998).

Line 18: EF stands for evaporative fraction. Please note that in order to improve the structure of our manuscript we do not discuss evaporative fraction in this section any longer. We use evaporative fraction as a secondary validation metric and mention it only briefly by referring to the newly created Supplement 3 from Section 3.3. We have mentioned in Section 2.1 that the full site names are given in current Figure 1 (was Figure 2 in the original manuscript). Writing out the site names all the time would increase the size of the manuscript considerably and would necessarily make labels in the figures unreadably small.

Line 19: Due to the complete rephrasing of the manuscript this is now discussed in the following way, which we hope clarifies:

"Scatterplots of normalised RMSE of latent heat flux (LE) and normalised RMSE of soil moisture for all sites are shown in Figure 9 and Figure 10 for the PS and CRNS two-objective calibration strategies

respectively. The RMSE values were normalised with respect to the default solutions, which therefore have normalised RMSE values of 1 (-). The black dots represent individual model runs and the default model run for each site is represented by a red cross The single-objective calibration solution of each single-objective calibration solution is shown as a blue triangle and the compromise solution of each two-objective calibration is shown as a green triangle. These two figures showed that the differences between the compromise solutions of the two calibration strategies observed for SR, DC, and UM in Figure 8, were less meaningful than analysis of Figure 8 only showed.

These findings are based on the shapes of the black point clouds in Figure 9 and Figure 10. We first look at the left edges of the black point clouds. When these edges are close to vertical, a small deterioration in RMSE-SM or RMSE-N (e.g. less than 0.05 (-)) would yield a large deterioration in normalised RMSE-LE (e.g. greater than 0.2 (-)). We observed this for three calibrations with PS soil moisture (SO, AR, and WR; indicated with pink line for WR) and in all these three cases, a less than 0.05 (-) change in normalised RMSE-SM would have yielded worse simulated latent heat flux than with the default parameter set. Seven calibrations with CRNS neutron counts (UM, DC, SO, KE, SR, CS, and WR; indicated with pink line for WR) showed a close to vertical edge. In four cases (UM, DC, KE, and CS) this would have yielded worse simulated latent heat flux than with the default parameter set. A negative slope for this side of the point cloud means an improvement in soil moisture estimation would mean a deterioration in latent heat flux estimation. We observed such negative slope for four calibrations with PS soil moisture (CS, MM, TR, and MO; indicated with continuous black line for MO) and for two calibrations with CRNS neutron counts."

Line 20: Please see our answer to the comment on line 19 for specification.

Line 21: Please see our answer to the comment on line 19. We rephrased this paragraph completely to better clarify.

Line 27 and line 28: Due to the rephrasing of the entire section we have removed the discussion on sites where strong coupling is expected. We did refer to sites in climatic transition zones, where soil moisture provides a first order control on evapotranspiration (Seneviratne et al. 2010). We here cite sentences summarising the findings from Section 3.4 of the revised manuscript:

"In summary, the two-objective calibrations against soil moisture (or neutron counts) and latent heat flux showed fewer substantial differences between calibration with PS soil moisture and calibration with CRNS neutron counts. These results indicate that the differences between both *single*-objective calibration strategies, which showed an advantage for CRNS observations, were possibly not as substantial as it seemed at first."

**COMMENT:**

Page 15 line...: 4: "day time only" Why suddenly "day time only"? Is this throughout the manuscript the case or just here? LE measurements are difficult in the night. Diurnal cycles are difficult to obtain. Only modeled LE at times of observed ET should be compared, because there is no observations at other times. I do not see this here. 15: "Weak coupling..." Can the forcing data be the reason here? I disagree, see comments to Figure. 16: "CRNS-N/LE" - rephrase and make clear. 20-21: Suggestions: Split sentence into two. 28: "root zone soil moisture" is where exactly and calculated how? 32: split sentence at "while" and remove "while". Also at other instances in the manuscript, this makes sence.

**ANSWER:**

Page 15 line 4: Latent heat flux and evaporative fraction performance were computed over day time values (Page 8 line 16 of the original manuscript) only in all analyses except when we plotted monthly mean diurnal cycles in Figure 9 of the original manuscript. The only instance when night time values were used was to plot the monthly mean diurnal cycles shown in Figure 9 (of the original manuscript). We have made this clearer by adding the following remark: "(night time data included

but not used for calibration as discussed previously)" to Section 3.3, page 12, lines 29-30 of the revised manuscript.

Line 15: Please note that this section was removed from the revised manuscript. Our discussions and conclusions changed and therefore the mentioning of the "weak coupling" was removed. Line 16: We now refer to the two different two-objective calibration approaches as: "calibrations with PS soil moisture" and "calibrations with CRNS soil moisture", within the context that makes clear we are talking about the two-objective calibrations.

Line 20-21: We have rephrased the sentence and moved it to Section 3.3 of the revised manuscript: "Another issue, which occurred for instance for the PS calibration at SO and for the CRNS calibrations at site AR, was that while surface energy flux estimation improved for a certain period, it deteriorated for another period (data not shown)."

Please note we removed the mentioning of site SR in this sentence because we now use latent heat flux as principle objective function. Latent heat flux yielded different improvement than evaporative fraction in this case.

Line 28: JULES does not provide a root zone soil moisture output weighted by the presence of roots that can be directly compared with the values of the wilting point and critical point soil moisture. Therefore we computed an estimation of root zone soil moisture by computing the soil moisture stress weighting factor for each layer at each time step (Equation 3). We then computed the relative contribution from each layer to the total root zone soil moisture stress factor. This relative contribution is a function of the root density and soil thickness. To finally obtain the weighted root zone soil moisture we multiplied the JULES soil moisture of each layer with the weighting factor. We now discuss the implications of self-adjusting behaviour of the wilting point and critical point soil moisture in Section 3.3.

Line 32: In our effort to reduce non-essential discussion in the manuscript we removed this sentence.

**COMMENT:**

Page 16 line ...: 4: Again the relatively weak coupling is discussed. Avoid repetition. Restructure. 5: "this suggests that how" - rephrase 9: Parameters should not move up or down after calibration. Variables and model states may be variable. "move closer" to what? 10: Which implications? Please name them and argue why. 14: "did not differ substantially". I disagree. They did differ. 17: "could be used"... you can put "should" instead. "we might have found worse fits". This is speculations. Please give reason or remove speculations. One could state: "You might have found better fits." 21: "Our findings support this." Your findings were that the JULES model is not an "improved land surface model", so the manuscript as is cannot support this. However, I see an impact of soil moisture states on latent heat flux in your model runs. Just have a look on Figure 11, how the RMSE in LE is reduced by calibration at sites DC,SO,KE,SR,WR. Only at few sites, RMSE in LE became higher. Soil moisture seemed to impact latent heat flux in JULES. The authors can be and should be more positive in the results, discussion and conclusion.

**ANSWER:**

Page 16 line 4: We agree with the reviewer that there are too many repetitions. We have restructured our manuscript to remove these. The discussion of the "weak coupling" was removed after reassessing our discussions and conclusions.

Line 5: We rephrased lines 4-6 and moved them to Section 3.3, page 12, lines 4-8: "One factor causing some of these limited improvements was that, when mean simulated root zone soil moisture (weighted with root density) increased after calibration, the values of the wilting point and critical point soil moisture parameters moved along (data not shown), yielding similar soil moisture stress. This happened for both calibration strategies at site KE, and for the calibration against CRNS neutron counts at sites MO and TR. This could relate to the limited value of simulated absolute soil

moisture for surface energy flux estimation in land surface models (Dirmeyer et al., 2000; Koster et al., 2009)."

Line 9: We have rephrased and split the sentence. We have moved this sentence to Section 3.3: "However, we also found the space between wilting point values and critical point values to decrease after calibration. This occurred when the standard deviation of the simulated soil moisture time series decreased due to calibration."

Line 10: We have removed this from the manuscript because it was non-essential information. Line 14: We think there was no substantial difference between calibrating against PS and against CRNS data with respect to latent heat flux. We completely revised our discussion of the twoobjective calibrations in Section 3.5 of the original manuscript (now Section 3.4) to make this clearer. We now start by showing differences between the compromise solutions of both two-objective calibration strategies were limited (Figure 12 of the original manuscript, now Figure 8). We then discuss the left edges of the point clouds in Figure 10 and Figure 11 of the original manuscript (now Figure 9 and Figure 10) and the lower edges of these point clouds. Based on the differences found for the single-objective calibrations and for the compromise solutions of the two-objective calibrations, we summarise in Section 3.4:

"In summary, the two-objective calibrations against soil moisture (or neutron counts) and latent heat flux showed fewer substantial differences between calibration with PS soil moisture and calibration with CRNS neutron counts. These results indicate that the differences between both *single*-objective calibration strategies, which showed an advantage for CRNS observations, were possibly not as substantial as it seemed at first."

**We also explain:**

"The spatio-temporal stability theory, which implies limited spatial variability in surface energy fluxes, could be one explanation for this (Vachaud et al., 1985; Teuling et al., 2006; Mittelbach and Seneviratne, 2012; Albertson and Montaldo, 2003). Another factor that has possibly played a role is that the spatial scale advantage of the CRNS was masked out by the possibly lower quality of its measurements (see also Section 3.1). Different hydrogen pools than soil moisture affect the neutron measurements at various temporal resolutions (e.g. rainfall interception; Baroni and Oswald, 2015). However, PS sensors also have their limitations. Different (electromagnetic) PS sensors have different designs and properties, which affect the data quality (Robinson et al., 2008; Blonquist et al., 2005). Soil type and soil specific calibration also affect the accuracy and precision of the PS data. For instance, the relationship between electrical permittivity and soil moisture content is strong for quartz rich soils, but less accurate for clay soils (e.g. Ishida et al., 2000; Robinson et al., 2008)."

Line 17: We have removed lines 16-18 because they are a repetition of page 11, lines 6-12. Those lines include the requested explanation:

"The RMSE values reduced relatively more for the CRNS calibration (70% on average over the twelve sites) than for the PS calibration (55% on average). The calibration method could possibly explain this. CRNS calibration was against observed neutron counts, while PS calibration was against observed neutron counts have an inverse relationship with soil moisture content, the PS calibration was possibly governed by avoiding larger errors occurring during a few brief soil moisture peaks. While focussing on getting the fitting for those peaks right, the PS calibration neglected the smaller errors during dry periods. This would then result in relatively smaller decrease in RMSE values than for the CRNS calibrations because those were fitted with heavier weights to the drier periods."

Line 21: We have rephrased this because our discussions and conclusions have changed. We refer to our answer to the comment on page 16, line 14. This sentence was rephrased and moved to the end of Section 3.3 of the revised manuscript:

"Previous research has also indicated that soil moisture alone is insufficient to estimate soil hydraulic parameters (Vereecken et al., 2008; 2015)."

**COMMENT:**

Table 2: Add symbols from the equations and use symbols within the text.

**ANSWER:**

Table 2: Equation symbols are now used in Table 2 and in the text body of the manuscript.

**COMMENT:**

Figure 1: I suggest removing the figure. The scale mismatch can be pointed out in a single sentence to save space for result figures e.g. The model grid cell size is 1km, the EC footprint is between 100m2 to 1km2 (put here diameter instead), the CRNS footprint is 300m in diameter, the PS soil moisture sensor measures few dm3.

**ANSWER:**

Figure 1: This figure has been removed. We think the scale mismatch is clear without further textual explanation. It is clear from page 2 lines 17-21 and page 3 lines 18-20 of the revised manuscript.

**COMMENT:**

*Figure 2: This Figure is very informative and necessary. Put Figure 2a and 2b instead of "upper".*

**ANSWER:**

Figure 2: Labels 2a and 2b have been included in the figure and we have referred to these as such in the figure caption.

**COMMENT:**

Figure 3: Informative, you may want to keep it, however consider removing.

**ANSWER:**

Figure 3: This figure has been moved to the newly created Supplement 1.

**COMMENT:**

Figure 4: Here, MO looks like it has a big discrepancy / bias in PS to CRNS soil moisture. Why is this the case / is this in the manuscript? Put the legend (PS, CRNS) beside or below the figure, not in the first subplot. Minimum of Y axis is missing.

**ANSWER:**

Figure 4: The clear difference between observed PS soil moisture and CRNS soil moisture retrieval shown in Figure 4 was discussed on page 9 line 4-5. The sites were put in order from small to large difference between the two soil moisture products in Figure 4 and in Figure 5. We discussed reasons for the differences between the two soil moisture products (observations/retrievals) in Section 3.1.2 and Section 3.1.3. We were unfortunately not able to identify why the two soil moisture products differed. We were not able to identify a specific reason for the large difference between the two soil moisture products at the Mozark (MO) site.

We have placed the legend below the figure and added minimum values at the vertical axes of the soil moisture plots. We have mentioned the abbreviation "SM" in the figure caption. We have also added the following sentence to the caption: "Daily precipitation is also shown here for each site."

**COMMENT:**

Figure 5: Informative. Keep it. CRNS observations are neutron flux. Soil moisture is retrieved from CRNS neutron flux observations. It should be rather something like CRNS soil moisture retrieval than CRNS soil moisture observation. Again, SM is abbreviated in your figure as SM, but not within the paper or at few instances. In general, abbreviations should be reduced.

**ANSWER:**

Figure 5: We respectfully disagree with the reviewer that CRNS soil moisture is less an observation than PS soil moisture. CRNS soil moisture values are obtained using a relationship (in our case COSMIC) with neutron counts, just like PS soil moisture values are obtained from a relationship with for instance electromagnetic wavelength. We will therefore keep the terminology as is. We have mentioned the abbreviation "SM" in the figure caption.

**COMMENT:**

*Figure 6: What is the added value to Figure 5, because JULES is calibrated based on the data used in Figure 5. I suggest to merge Figures into one with e.g. Figure 5a and 5b.*

**ANSWER:**

Figure 6: The authors would like to argue that there is added value of Figure 6 to Figure 5 (numbering in the original manuscript). Figure 5 shows how similar or different the two soil moisture products are at the different sites. Figure 6 shows how successful calibration was, i.e. how much the Root Mean Squared Error between simulated and observed/retrieved soil moisture was reduced by calibration. Calibration was not as successful in all instances. The relative improvement in Root Mean Squared Error differed between calibration against PS soil moisture and CRNS soil moisture retrieval and also differed between sites. We have decided not to combine Figure 6 with Figure 5 because we think this would increase confusion between their respective purposes (data analyses, versus effect of calibration). Moreover, they are discussed in two separate section.

**COMMENT:**

Figure 7:Add grid lines. How can CRNS and PS soil moisture at MO be so different as in Figure 5, but Model results after calibration be so similar as in Figure 7. It seems very strange. Also you plot a 3 year time series with hourly values. It will be much easier to read, with more information and will have even more meaning if you average over days or months. So far, it is a lot of variability, which is clear beforehand if hourly values are plotted over the course of 3 years. You may consider merging Figure 7 and Figure 4.

**ANSWER:**

Please notice that the overall calibration based on PS and CRNS (i.e., top and bottom MO panels in Figure 7) actually yielded different results. The Figure b of this answer combines both plots to highlight this fact (notice default run line is omitted as it is irrelevant to this comparison). We decided to keep hourly values because they represent the soil moisture dynamics, which are important with respect to surface energy flux simulation (e.g. Teuling et al., 2009). We have added grid lines.

---

## Referee Report (RR1)

The topic is of significant interest to the HESS readers as it combines advanced multi-scale measurements with land surface modeling. The manuscript is fluently written and represents a much needed study to advance hyper-resolution land surface models through calibration with scale-consistent terrestrial soil moisture observations using cosmic ray neutron sensors.

The authors have clearly addressed the points I have mentioned in the previous review and provided arguments for their case where appropriate. The manuscript improved meaningfully and should be published in HESS after minor revision. Please address the comments below.

**General:**

Figure 6: Based on your Figure 6, I actually find that CRNS do improve modeled LE in most of the cases shown. Furthermore, pretty much in contrast to PS measurements. This is a contradiction to the conclusion and actually would strengthen the publication.

I suggest splitting Figure 6 into two subfigures. You will recognize two point clouds:

> 1 cloud for the PS which covers the entire area (=> no improvement to LE, but to SM)

> 1 cloud for the CRNS which covers the better left corner (=> improved LE+SM)

The centers-of-mass for PS and CRNS points are different. This is not adequately reflected in the text. I would argue that your one-objective calibration with the CRNS improves LE overall (not at all sites, but overall), while PS data does not. This would be an excellent result for the present study but it is not drawn / concluded yet. It would be an example of a successful scale consistent calibration compared to the scale mismatch calibration with PS.

The finding that CRNS improve LE overall actually contradict to a few of the results and conclusion in the manuscript as is:

p11 l22:"While calibration errors decreased for soil moisture, latent heat flux estimation improved for fourteen out of twenty-four calibrations (Figure 6). This means that an improvement in simulated soil moisture did not necessarily lead to better estimation of surface energy fluxes."

p15 l15: "The single-objective calibration of JULES parameters against point-scale soil moisture and Cosmic-Ray Neutron Sensor neutron counts did not necessarily yield an improvement in latent heat flux simulation. The analysis of these single-objective calibrations and multi-objective calibrations (against (1) PS soil moisture and latent heat flux and (2) CRNS neutron counts and latent heat flux) revealed that differences between calibrations with these two soil moisture observation methods did overall not yield substantially different surface energy flux estimations."

Accordingly, few sentences of the results, conclusion and abstract need to be rewritten to include the new results. Furthermore I suggest to combine the new Figure 6 a+b with Figure 8 a+b (also split this one into PS + CRNS).

There are some cases in which consistency throughout the manuscript would be beneficial e.g. RMSE or RMSD, always use "energy fluxes", write "eddy" with or without capital letter.

**Minor points:**

p1 l16-19: I still see a beneficial, however limited effect on latent heat flux from the calibration of soil moisture. Quantify the even limited (but this is subjective) effect of calibrating soil parameters with the si in the abstract.

p1 l.27: „Koster et al. 2004"

p2 l.6 (Blyth et al. 1993) – is there a newer reference available, as land surface models and stomatal resistance parameterization should have improved significantly from 1993.

p3.l10: The hypothesis could be more clear and better linked to the question two lines before. I suggest to reuse "reduced observation scale" and "LSM energy flux". I would also omit "estimates", because it is rather limiting and not needed.

p.3 l 24-25: sentence seems out of place. Consider moving or removing.

p4 l 9: Many readers familiar with the topic will skip reading the introduction and may want to start reading here. Therefore, I suggest here to write "Point Scale (PS)" out once more.

p5 l6 & 9: In case it is, add "linear correlation" to be more clear.

p5 l 11: m3 m-3 (add minus)

p7 l25: remove "observed" in "observed PS observed";

p7 l25: "CRNS observed neutron counts" shall be consistent throughout the manuscript. Sometimes it is CRNS neutron counts, sometimes CRNS soil moisture. I suggest to use "CRNS neutron counts" here and later use more often the simple and short version "CRNS" instead.

p7 l28: is RMSE the same as RMSD? Better be consistent and use one single term.

p7 l30 "parameter: b" – b in brackets

p8 l6: here and in l15, Pedotransfer function either with capital letters or not. Either is fine, but use consistently throughout the manuscript.

p8 l10 "Clay" +l

p8 l12: If possible use the greek symbol for bulk density rho and "dry" in subscript.

p9 l24: remove ".."

l25: add space before "Figure 3"

l26: "between PS and CRNS soil moisture (…"

l26 "bias" – I suggest to use "bias" in the Figure as well or use the same wording in the text as was used in the Figure.

l28-29: The meaning of the sentence is not 100 percent clear to me.

p10 l6: AmeriFlux with capital F, and same throughout the manuscript (see http://ameriflux.lbl.gov/ website)

p10 l13: There are more factors to neutron intensity e.g. altitude, sensor type. Those should be named as well.

p10 l17: "quite similar" is unclear. Rather "close by RMSE …" or showed "similar dynamics but different means"

p10 l17 remove ") "

p9 l25-p10l25: I suggest to give the paragraph a more concise and maybe shorter structure.

p10 l33: clarify or omit "(bias)".

p11 l9: Start a new sentences with "Actually…"

l12: add space after Figure 5

l29: Is there one space too much between "flux  estimation"?

l30-32: Reverse sentence order: "Improvement in latent heat flux was actually substantial in jfour cases for PS calibration and five cases for CRNS calibration.". remove "just"

l32-33: Same, reverse order.

p12 l6: "Happened". Maybe reformulate to "was the case".

l20: Remove "only"

l21 Add space bettwen wasbelow

l30: I suggest to remove "as discussed previously".

l34: "AT" -> "At"

p13 l11: "sites. To"

l21: Error! Reference source not found

l22: I suggest to remove "values". I also suggest to replace "which therefore have normalized RMSE of 1" by "to 1 (-)"

p13 l23: "cross. The"

p15 l5: The first sentence should well introduce the conclusion but the JULES LSM is not mentioned here while being the second most important component of your study. Add JULES to what you investigated and leave something else for the second sentence.

p15 l11: "happened in different ways"... rephrase. Differences do not happen. They are there and have reasons.

l14: no new paragraph here

l19: Suggestions: new paragraph before "These"

l19-21: I have a problem understanding the sentence: The "did not... to reject...reduce...mismatch does not..." makes it very hard for me to understand what this sentence actually wants to say. I suggest to rephrase into the direction: "outcomes did... to confirm... improve scale consistency...".

l22: Here (in the conclusion) it is important to be precise. Energy flux estimates can come from Penman-Monteith, measurements or models (or even laboratories etc.). I suggest to go through the manuscript and use e.g. "prediction" or "simulated" (as used in your the abstract) when the energy flux is modeled, and "observation" when it is measured by eddy covariance.

l22:  Add "energy" between "surface flux". Be consistent throughout the manuscript, not sure where else it is used this way.

l23: remove "relatively"

l23: replace against with "with"

l24: remove "simulated surface energy fluxes".

l25: If I remember it correctly, the model was calibrated with "CRNS neutron flux", not with "CRNS soil moisture data". There may be other occurrences in the manuscript.

l25: remove "Another factor, related to this and that contributed to this result, was that".

p16 l5: Rephrase sentence, not clear yet.

l6: remove "."

Figure 5:  Why is the calibrated soil moisture in CRNS SM at WR site (lower part) so much noisier than the calibrated soil moisture at PS SM at WR site?

p37: Move or remove "Appendix 3:"

p38: "We wide" rephrase.

---

## Author Response (AR2)

**Final answer to review comments on manuscript originally entitled "Land surface model performance1 using cosmic-ray and point scale soil moisture measurements for calibration" by Iwema et al. submitted to Hydrology and Earth System Sciences**

We thank the Editor and the reviewers for the positive comments and suggestions that significantly added to the quality of the manuscript. We address the comments of Referee #1 Roland Baatz below. The two comments by Anonymous Referee #3 (one figure reference and the title "Appendix 3") have been addressed in the revised manuscript. At the end of this answer a marked up version of the revised manuscript is included.

Sincerely,

Joost Iwema and co-authors

**Answer to comment by Referee #1 Roland Baatz**

We would like to thank the Referee for carefully reviewing the manuscript. A point-by-point reply to the comments is provided below with original comments shown in Italic font.

*The topic is of significant interest to the HESS readers as it combines advanced multi-scale measurements with land surface modeling. The manuscript is fluently written and represents a much needed study to advance hyper-resolution land surface models through calibration with scale-consistent terrestrial soil moisture observations using cosmic ray neutron sensors.*
*The authors have clearly addressed the points I have mentioned in the previous review and provided arguments for their case where appropriate. The manuscript improved meaningfully and should be published in HESS after minor revision. Please address the comments below.*

**ANSWER:**
We thank the referee his encouraging comments. We have addressed the newly raised issues below.

*GENERAL:*
**COMMENT:** *Figure 6: Based on your Figure 6, I actually find that CRNS do improve modeled LE in most of the cases shown. Furthermore, pretty much in contrast to PS measurements. This is a contradiction to the conclusion and actually would strengthen the publication.*
*I suggest splitting Figure 6 into two subfigures. You will recognize two point clouds:*
> *1 cloud for the PS which covers the entire area (=> no improvement to LE, but to SM)*
> *1 cloud for the CRNS which covers the better left corner (=> improved LE+SM)*
*The centers-of-mass for PS and CRNS points are different. This is not adequately reflected in the text. I would argue that your one-objective calibration with the CRNS improves LE overall (not at all sites, but overall), while PS data does not. This would be an excellent result for the present study but it is not drawn / concluded yet. It would be an example of a successful scale consistent calibration compared to the scale mismatch calibration with PS.*

**ANSWER:** We thank the reviewer for this interesting suggestion. However, we believe it is easier to compare the two calibration strategies when both are presented within the same subplot. With respect to the centres of mass, it is also easier to compare these when the two strategies are presented within the same plot. We have included the following sentence on lines 26-29 of page 11 of the revised manuscript to mention the centres of mass:

"The differences between the two calibration strategies can also be seen from the different locations of the centres-of-mass of the two point clouds. The cloud of the CRNS strategy is clearly located more to the lower left corner in **Error! Reference source not found.**."

We mentioned in the manuscript that the single-objective calibrations with PS data did yield more improvement in latent heat flux than the single-objective calibrations with CRNS data (lines 3-4 of page 13). However, based on the results of the two-objective calibrations, we concluded these differences were not as substantial as seemed at first. To amplify the difference between the single-objective calibration results and the results from the two-objective calibration, we changed in the Conclusions (page 15, lines 14-18):

"The single-objective calibration of JULES parameters against Cosmic-Ray Neutron Sensor neutron counts yielded better simulated latent heat flux than single-objective calibration against point-scale soil moisture. The analysis of multi-objective calibrations (against (1) PS soil moisture and latent heat flux and (2) CRNS neutron counts and latent heat flux) however revealed that differences between calibrations with these two soil moisture observation methods did overall not yield substantially different surface energy flux estimations."

And we changed the following sentence in the abstract (page 1, lines 16-18):

"Calibrations against the two soil moisture products alone did show an advantage for the Cosmic-Ray technique. However, further analyses of two-objective calibrations with soil moisture and latent heat flux showed no substantial differences between both calibration strategies."

**COMMENT:** *The finding that CRNS improve LE overall actually contradict to a few of the results and conclusion in the manuscript as is:*

*p11 l22:"While calibration errors decreased for soil moisture, latent heat flux estimation improved for fourteen out of twenty-four calibrations (Figure 6). This means that an improvement in simulated soil moisture did not necessarily lead to better estimation of surface energy fluxes."*

*p15 l15: "The single-objective calibration of JULES parameters against point-scale soil moisture and Cosmic-Ray Neutron Sensor neutron counts did not necessarily yield an improvement in latent heat flux simulation. The analysis of these single-objective calibrations and multi-objective calibrations (against (1) PS soil moisture and latent heat flux and (2) CRNS neutron counts and latent heat flux) revealed that differences between calibrations with these two soil moisture observation methods did overall not yield substantially different surface energy flux estimations."*

**ANSWER:** Note that we used "not necessarily" on p11 L19 of the previous version of the manuscript, which reflects the fact that CRNS calibration did not improve LE in all cases.

Please see our answer to the previous comment for how we modified the sentence on p15, l15.

**COMMENT:** *Accordingly, few sentences of the results, conclusion and abstract need to be rewritten to include the new results. Furthermore I suggest to combine the new Figure 6 a+b with Figure 8 a+b (also split this one into PS + CRNS).*

**ANSWER:** We have updated the Conclusions and abstract sections accordingly as mentioned in our answers to the previous comments. Note that one of the key points in our paper is the impact of

carrying out a two-objective calibration on model performance, in comparison with a single-objective calibration. This is done by presenting their results separately. Hence, we have decided to keep Figure 8 unchanged in accordance to our decision about Figure 6 (explained above).

**COMMENT:** *There are some cases in which consistency throughout the manuscript would be beneficial e.g. RMSE or RMSD, always use "energy fluxes", write "eddy" with or without capital letter.*

**ANSWER:** We use RMSD when we compare two observations and RMSE when we compare observations with simulations. We have updated Figure 3 for consistency.

***MINOR COMMENTS***
**We thank the reviewer for the minor comments provided. We took all comments into account. The changes in the manuscript are highlighted in the marked-up version of the manuscript. Here below we focus on the comments that required a more detailed response only.**

**COMMENT:** *p1 l16-19: I still see a beneficial, however limited effect on latent heat flux from the calibration of soil moisture. Quantify the even limited (but this is subjective) effect of calibrating soil parameters with the si in the abstract.*

**ANSWER:** Note we do not go in detail on how much actual improvement is observed because the key point here is to discuss the impact (in this case, lack of model improvement) when going from single-objective to two-objective calibration, suggesting a limited effect of calibrated soil parameters on soil moisture dynamics and surface fluxes, simultaneously.

**COMMENT:** *p2 l.6 (Blyth et al. 1993) – is there a newer reference available, as land surface models and stomatal resistance parameterization should have improved significantly from 1993.*

**ANSWER:** Please note that we use this reference as an example to support our point. However, we would like to keep this reference as is to show the knowledge that stomatal resistance measured at leaf level is not the same as canopy stomatal resistance has been existing for a while now.

**COMMENT:** *Figure 5: Why is the calibrated soil moisture in CRNS SM at WR site (lower part) so much noisier than the calibrated soil moisture at PS SM at WR site?*

**ANSWER:** The soil moisture after calibration with CRNS neutron counts at WR is much noisier than after calibration with PS soil moisture because the CRNS observations are much noisier due to inherent uncertainties, as explained on lines 21-23 of page 9 of the new manuscript. At the WR site this effect is especially strong due to the relatively low neutron count and possibly due to temporal changes in other neutron mitigating factors. We have included a sentence in Section 3.2 of the new version of the manuscript explaining this specifically for site WR.

[revised manuscript text omitted]

---

## Author Response (AR3)

**Final technical corrections of manuscript entitled "Land surface model performance using cosmic-ray and point scale soil moisture measurements for calibration" by Iwema et al. submitted to Hydrology and Earth System Sciences**

We thank the Editor for the positive comments and suggestions that significantly added to the quality of the manuscript. We address the comments below. At the end of this answer a marked-up version of the revised manuscript is included.

Sincerely,

Joost Iwema and co-authors

**Answer to comments by Editor Harrie-Jan Hendricks-Franssen**

**Thanks for the revised version of "Land surface model performance using cosmic-ray and point scale soil moisture measurements for calibration". I reviewed your revised version and found that it is suitable for publication now after handling some technical corrections:**

ANSWER: We thank the reviewer for his positive comments and acceptance of our manuscript for publication in HESS.

**COMMENT: P1, L8: I think "very high resolution" instead of "hyperresolution" is better for this scale.**

ANSWER: We have changed "hyperresolution" to "very high resolution".

**COMMENT: P5, L1: "California Climate Gradient Sites" instead of "California Climate Gradients Sites"? Here and elsewhere in the paper.**
ANSWER: We have changed "California Climate Gradients Sites" to "California Climate Gradient Sites".

**COMMENT: P5, L29: It would be good to indicate in the paper how frequent and large were the data gaps.**
ANSWER: We have included the following sentences on page 14 line 29 to page 15 line 6, to clarify the number of gaps filled with NLDAS and Santa Rita data:

*"Gaps larger than 30 days were filled with NLDAS-2 data at sites WR, CS, and SO. At site WR a single gap of 110 days in shortwave and longwave radiation was filled. At site SO a gap of 66 days in atmospheric pressure was filled with NLDAS-2 data. At site CS two gaps of 200 days in atmospheric pressure was filled and the entire time series of air temperature was filled with a linear relationship based on data from four preceding years. At site SR, gaps before applying the moving window procedure varied between 3% and 15 % of the time for precipitation and wind speed, respectively and atmospheric humidity was missing for 35% of the time. These gaps were mostly filled with a linear relationship with data from the Sahuarita site, located approximately 14 km to the north-west from the SR site, except for downward longwave radiation. Remaining gaps were filled with the moving window procedure."*

**COMMENT: P7, L2: layer i; i should be italic.**
ANSWER: We made "i" italic.

**COMMENT: P7, L20: comma missing.**
ANSWER: We inserted the comma.

**COMMENT: P7, L29: closing bracket missing.**
ANSWER: We inserted the closing bracket.

**COMMENT: P13, L15-L17: reformulate sentence. Following sentence also unclear in relation to sentence before.**
ANSWER: We have reformulated these sentences:

*"The results from the two-objective calibrations suggest that latent heat flux estimation improvement with respect to the compromise solutions (%-change RMSE-LE, vertical axis in Figure 8) was similar for both two-objective calibration strategies (Figure 8). Only at three sites was the difference in improvement (%-change RMSE-LE, vertical axis in Figure 8) between the two-objective calibration strategies more than 5% (at sites SR, DC, and UM)."*.

**COMMENT: P13, L25: "show" instead of "showed".**
ANSWER: We changed to "show".

**COMMENT: P13, L26-L27: very unclear.**
ANSWER: We have reformulated:

*"Figure 9 and Figure 10 show that the substantial (i.e. more than 5%) differences between the two two-objective calibration strategies in improvement of simulated latent heat flux observed for SR, DC, and UM in Figure 8, were less meaningful than seemed initially from analysis of Figure 8."*

**COMMENT: P14, L15: fewer than what? Specify.**
ANSWER: We have rephrased:

*"In summary, the two-objective calibrations against soil moisture (or neutron counts) together with latent heat flux showed, compared to the single-objective calibrations, fewer substantial differences between calibration with PS soil moisture and calibration with CRNS neutron counts."*

**COMMENT: P15, L11: you could add that these differences were however small.**
ANSWER: We rephrased:

*"These analyses showed the differences between PS and CRNS soil moisture varied in nature at the investigated sites but such differences were however small."*

**COMMENT: Figure 1: Ticks on vertical axes.**

ANSWER: We inserted ticks on vertical axes.

**COMMENT: Figure 7: Repetition in caption. Correct.**

ANSWER: We removed the repetition.

**COMMENT: Figure 9: Include ticks on axes.**

ANSWER: We have made the ticks on the axes in Figure 9 and Figure 10 larger to make them better visible.

**COMMENT:  Additional thoughts:**

ANSWER: We address these comments hereinafter.

**COMMENT: - What is effective sampling depth of CRNS and how does it differ from PS at the different sites? What about difference in temporal resolution of observations? Could these differences play a role?**

ANSWER: The effective sampling depth of the CRNS varied from about 5 to 40 centimetres and differed between sites and over time due to changing water content. The PS sensor depths were however constant in time and varied from just a few centimetres below the surface to 30 centimetres. In general, the sensing depths of both techniques were therefore similar, but differences did occur. At Santa Rita Creosote (SR), for example, the CRNS sensing depth was relatively deep (20-40 centimetres) while the PS sensors used were installed at 2.5 and 12.5 centimetres. In that case, the CRNS had greater depth. At site Morgan Monroe, the PS sensors covered a layer from the surface to 30 cm depth, whereas the CRNS sensing depth was 5-20 centimetres. The temporal resolution of the two techniques differed, with PS soil moisture integrated to hourly intervals whereas for CRNS data a 5-hours moving average filter was applied. Comparison of hourly and daily averaged PS soil moisture data (not shown) did however reveal minor differences throughout the time series. The same was found for a comparison between CRNS data with a 5-hours moving average applied and daily averaged CRNS data. Additionally, data comparison between daily PS and CRNS soil moisture data yielded results like the comparison of hourly PS and 5-hour moving average CRNS soil moisture presented in the manuscript. These results suggested the difference in temporal variability within a day was minimal across sites. Exceptions occurred for high peaks after intense rainfall, which yielded higher peaks in hourly soil moisture than in daily soil moisture.

**COMMENT: - How are you sure that for gaps in precipitation time series the precipitation was zero?**

ANSWER: We cannot be sure that for gaps in precipitation time series the precipitation was zero. We chose to set missing precipitation values to zero because gap-filling by interpolating data would generate even greater uncertainty by the addition of random precipitation amounts, affecting the water balance even more. Filling the gaps with linear relationships with for instance NLDAS-2 data

was also not a wise idea because correlations were very weak (not shown). Gaps in precipitation data are a challenge in modelling exercises like these.

**COMMENT: - I think that indeed the role of deeper roots can be important and including it in a future analysis would be important.**

ANSWER: We agree with the Editor that it would be important to include the role of deeper roots in future analysis. This issue was also raised by Referee Ryan Teuling and we answered that such analysis was beyond the scope of our research. Including such analysis in future research might be easiest if a limited number of sites with accurate site information is used.

**COMMENT: - What happens if you would use just one PS sensor? Then you have really a scale mismatch. A large number of PS sensors approaches the soil moisture content for the footprint.**

ANSWER: At six of our twelve sites one profile (with one or two PS sensors) was used only. At two sites, two profiles were used and at the four other sites four or six profiles were used (Fig.A1.1). The number of PS sensors per profile was one at eight sites and two at four sites. We did not find necessarily larger differences between the two soil moisture sensing techniques at sites with more PS profiles. Actually, the sites with the smallest difference between the two techniques (UMBS) just one profile was used. On the other hand, sites with more than two profiles were among the seven sites with most similar soil moisture between the two sensing techniques. In our case, it was not possible to use a single profile at all sites because the data available from Ameriflux sites with multiple PS profiles were averages of these multiple profiles. A more detailed investigation at sites with separable PS profile data could possibly reveal more specific differences and could be an interesting future study.

[revised manuscript text omitted]